# Near infrared-activatable biomimetic nanogels enabling deep tumor drug penetration inhibit orthotopic glioblastoma

Dongya Zhang[1], Sidan Tian[2], Yanjie Liu[1], Meng Zheng[1], Xiangliang Yang [2], Yan Zou [1,3] ✉, Bingyang Shi [1,3] ✉ & Liang Luo [2,4] ✉

Glioblastoma multiforme (GBM) is one of the most fatal malignancies due to the existence of blood-brain barrier (BBB) and the difficulty to maintain an effective drug accumulation in deep GBM lesions. Here we present a biomimetic nanogel system that can be precisely activated by near infrared (NIR) irradiation to achieve BBB crossing and deep tumor penetration of drugs. Synthesized by crosslinking pullulan and poly(deca-4,6-diynedioic acid) (PDDA) and loaded with temozolomide and indocyanine green (ICG), the nanogels are inert to endogenous oxidative conditions but can be selectively disintegrated by ICG-generated reactive oxygen species upon NIR irradiation. Camouflaging the nanogels with apolipoprotein E peptide-decorated erythrocyte membrane further allows prolonged blood circulation and active tumor targeting. The precisely controlled NIR irradiation on tumor lesions excites ICG and deforms the cumulated nanogels to trigger burst drug release for facilitated BBB permeation and infiltration into distal tumor cells. These NIR-activatable biomimetic nanogels suppress the tumor growth in orthotopic GBM and GBM stem cells-bearing mouse models with significantly extended survival.

Glioblastoma multiforme (GBM) is the most common primary tumor in the central nervous system, accounting for about 40% of the total intracranial malignant tumor incidence[1,2]. According to the World Health Organization classification, GBM is of grade IV histological malignancy and the median survival of glioblastoma patients is only about 12 months[3,4]. Although a variety of modalities, including surgery[5], radiotherapy[6], chemotherapy[7], and other emerging methods such as photodynamic therapy (PDT)[8,9], have been developed for GBM treatment, their performances are far below expectations with limited survival. In particular, to approach effective drug concentrations in the GBM lesions, which is highly desired for optimal therapeutic outcomes, suffers from short blood circulation, limited blood-brain barrier (BBB) penetration, and insufficient tumor uptake[10–12]. Efficient delivery of intravenously administered therapeutics across the BBB and other biological barriers into the tumor remains an insurmountable challenge for GBM treatment.

Delivering therapeutics through engineered nanomaterials, or the so-called nanomedicines, has been considered revolutionary strategies in overcoming a series of biological barriers for anticancer treatment[13,14], and also offers an opportunity to integrate multiple treatment modalities[15–17]. However, to assert effective drug accumulation in GBM is still difficult, mostly because of inefficient drug extravasation and

[1]Henan-Macquarie Uni Joint Centre for Biomedical Innovation, Academy for Advanced Interdisciplinary Studies, Henan Key Laboratory of Brain Targeted Bio-nanomedicine, School of Life Sciences, Henan University, 475004 Kaifeng, Henan, China. [2]National Engineering Research Center for Nanomedicine, College of Life Science and Technology, Huazhong University of Science and Technology, 430074 Wuhan, China. [3]Macquarie Medical School, Faculty of Medicine, Health and Human Sciences, Macquarie University, Sydney, NSW 2109, Australia. [4]Key Laboratory of Molecular Biophysics of the Ministry of Education, College of Life Science and Technology, Huazhong University of Science and Technology, 430074 Wuhan, China. ✉e-mail: yzou@henu.edu.cn; bs@henu.edu.cn; liangluo@hust.edu.cn

penetration in brain tumors[18], as well as the failure of tumor-specific drug release[19,20]. Although active ligands such as apolipoprotein E (ApoE) peptide can facilitate BBB crossing and tumor uptake of nanomedicines through specific ligand-receptor interactions[21,22], infiltration from blood vessels into GBM tumor tissue and to distal tumor cells remains largely inefficient for bulk nanoparticles. In addition, many nanocarriers responding to pH values, redox conditions, and tumor overexpressed enzymes have been developed for tumor-specific drug release[23–27]. However, these stimulations based on tumor microenvironment (TME) are not readily applicable to nanomedicines in blood vessels. Even in the tumor regions, the TME that benefits the desired drug release is usually 100–200 μm away from the tumor blood vessel networks[26,28,29]. Moreover, passive drug release based on TME stimulation can be misled by non-tumor conditions similar to TME[30,31]. For instance, reactive oxygen species (ROS)-responsive carriers represent a very promising category in nanomedicine, since most tumor tissues have shown higher ROS levels than normal[32–34]. However, inflammation or amyloid deposits in brain also induces high endogenous ROS levels[35–38], which can easily alter the designated drug release in GBM tumors and affect the therapeutic outcomes. Precisely activatable drug delivery nanoplatforms are therefore highly desired to warrant efficient BBB penetration and tumor-specific release of therapeutic agents for successful GBM treatment.

In this work, we have created a nanogel system by crosslinking pullulan and an oxidatively degradable conjugated polymer poly(deca-4,6-diynedioic acid), or PDDA[39] as shown in Fig. 1. PDDA is inert to endogenous oxidative conditions, but can degrade in the presence of ROS generated by photosensitizers upon light irradiation[40]. Loading both indocyanine green (ICG), an FDA-approved near-infrared (NIR) photosensitizer, and temozolomide (TMZ), the first-line chemotherapeutic drug for GBM, into the nanogels yields NGs@TMZ/ICG, which are then camouflaged with ApoE peptide-decorated erythrocyte membranes for prolonged blood circulation and active tumor targeting. The enhanced stability of finally formed ARNGs@TMZ/ICG in physiological conditions allows them to cumulate efficiently in GBM lesions after intravenous administration. NIR light is then applied manually to activate ARNGs@TMZ/ICG when they reach maximal accumulation in GBM lesions. ICG can generate ROS to deform the nanogels and trigger the burst release of both TMZ and ICG for facilitated extravasation and deep tumor penetration (Fig. 1). The accumulated TMZ and ICG in deep GBM lesions hence boost the efficacy of combined photodynamic-chemotherapy to alleviate orthotopic GBM tumors in mouse models, demonstrating that this NIR-activatable biomimetic nanogel system as a clinical solution for brain tumor treatment.

## Results

### Synthesis and NIR-induced disintegration of NGs@TMZ/ICG

The NGs@TMZ/ICG nanogels are readily fabricated via two steps, the crosslinking between PDDA and pullulan and the co-loading of TMZ and ICG. As schematically illustrated in Fig. 2a, when 808 nm NIR irradiation

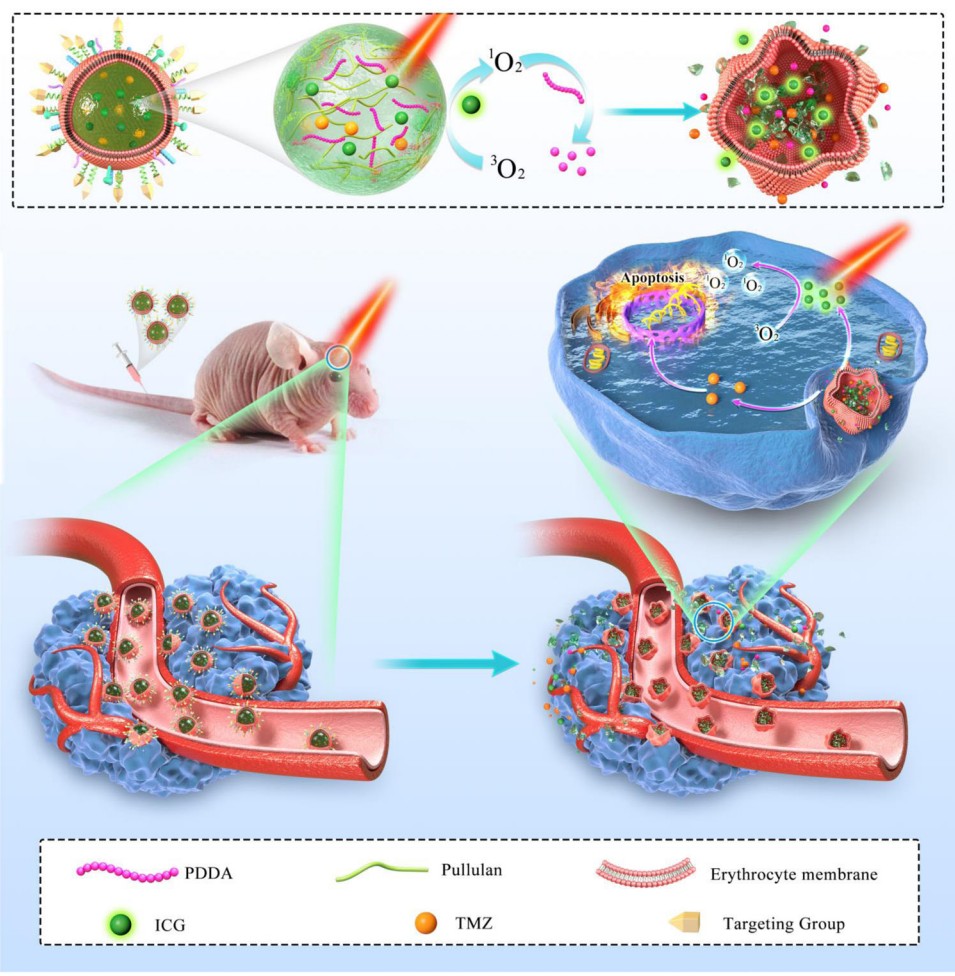

**Fig. 1 | Schematic illustration of near infrared-activatable biomimetic nanogels for orthotopic GBM therapy.** ApoE-modified erythrocyte membranes were cloaked on the surface of near infrared-activatable biomimetic nanogels co-loaded with photosensitizers ICG and TMZ. These biomimetic nanogels accumulate effectively in GBM lesions after the intravenous injection in GBM bearing mice. NIR irradiation triggers the release of TMZ and ICG to accumulate into the deep lesions of the tumor, resulting in effective photodynamic and chemotherapy for GBM tumor cell inhibition.

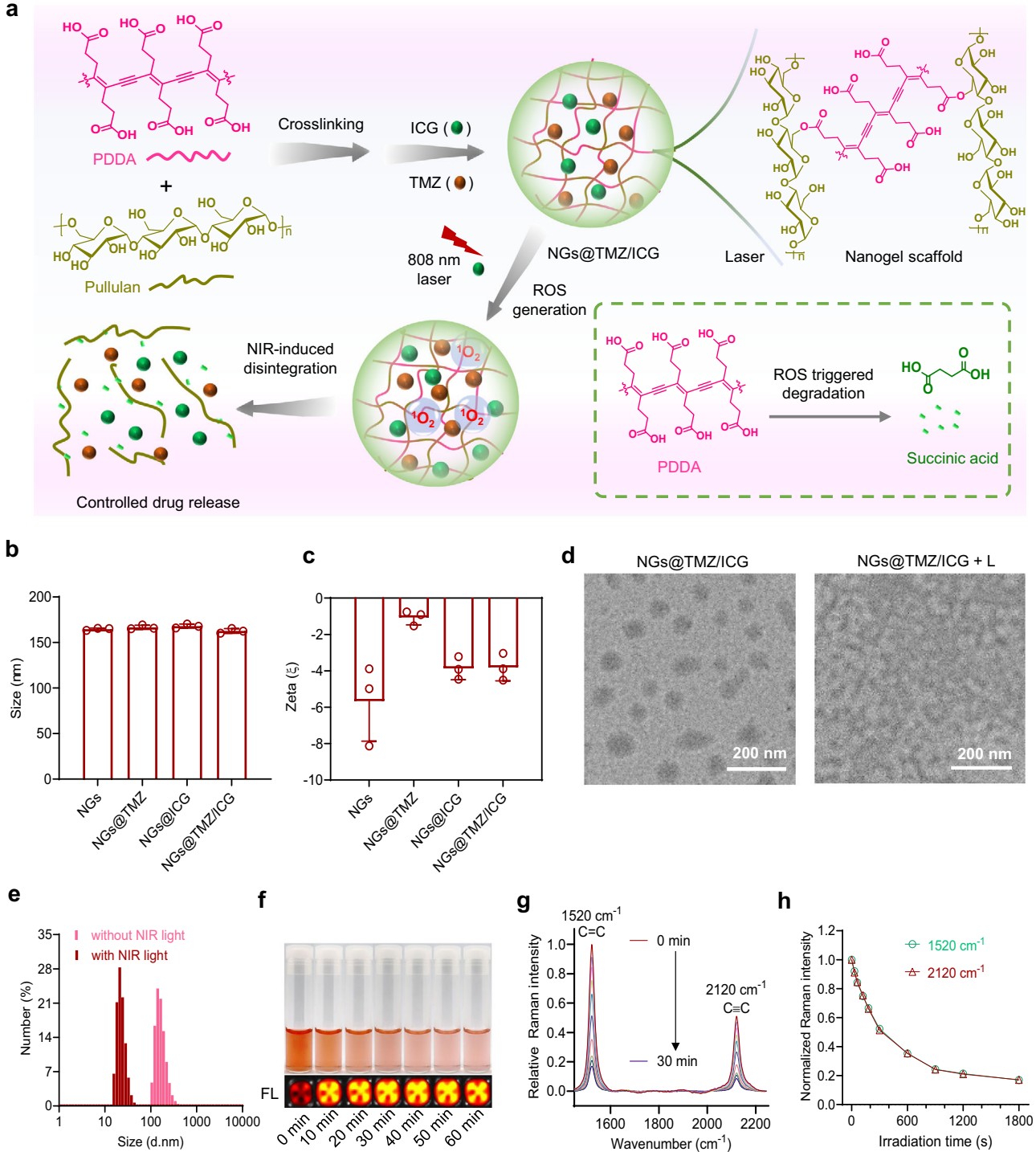

**Fig. 2 | Preparation and NIR-induced disintegration of NGs@TMZ/ICG.**
**a** Schematic illustration of the synthesis of the nanogels and their NIR-induced disintegration. **b**, **c** Size distribution (**b**) and zeta potential (**c**) of the corresponding nanoparticles measured by DLS (n = 3 independent experiments). Data are presented as mean ± SD. **d** TEM images of the NGs@TMZ/ICG before and after NIR irradiation (808 nm, 0.5 W cm⁻², 5 min). **e** Particle sizes of NGs@TMZ/ICG before and after NIR irradiation (808 nm, 0.5 W cm⁻², 5 min) measured by DLS. **f** Photographs and the florescence images (FL) of the RB-containing nanogels upon light irradiation for different time (520 nm, 5 mW cm⁻²). **g** Raman spectra of NGs@TMZ/ICG after exposure to NIR irradiation for different time (808 nm, 0.5 W cm⁻²). **h** Change in the normalized Raman intensity of C=C bonds (1520 cm⁻¹) and C ≡ C bonds (2120 cm⁻¹) as a function of sunlight irradiation time.

is applied to NGs@TMZ/ICG, ICG generates ROS to degrade PDDA into succinic acid, a biocompatible small molecule that naturally occurs in organisms. The nanogels are therefore disintegrated to release TMZ and ICG accordingly. Dynamic light scattering (DLS) measurement showed that all prepared nanoparticles, including NGs@TMZ/ICG, single agent loaded nanogels NGs@TMZ and NGs@ICG, and blank nanogels

containing no drug, showed similar hydrodynamic sizes of around 170 nm (Fig. 2b) and slightly negative surface charges (Fig. 2c). The transmission electron microscope (TEM) image of NGs@TMZ/ICG displayed their sphere morphologies with less than 200 nm in diameter. After NIR irradiation, the nanoparticles exhibited apparent disintegration with decreased sizes and indistinct shapes ("NGs@TMZ/ICG + L" in

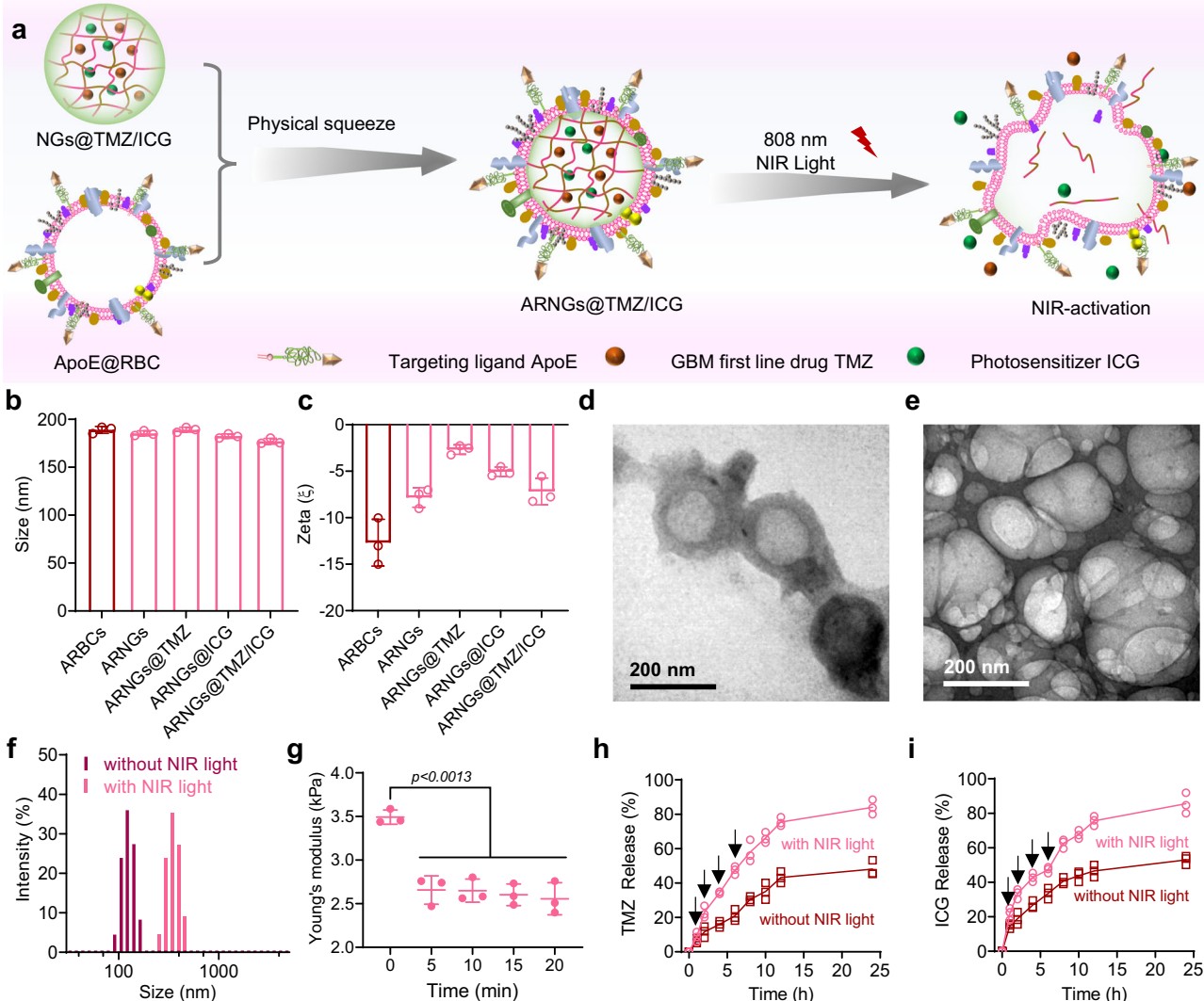

**Fig. 3 | Preparation of ARNGs@TMZ/ICG and NIR-induced deformation.**
**a** Schematic illustration of camouflaging ARNGs@TMZ/ICG by ApoE peptide-decorated erythrocyte membranes and NIR-induced deformation of the nanogels. **b**, **c** Size distribution (**b**) and zeta potential (**c**) of various nanoparticles measured by DLS (n = 3 independent experiments). Data are presented as mean ± SD **d**, **e** TEM images of ARNGs@TMZ/ICG before (**d**) and after (**e**) NIR irradiation (808 nm, 0.5 W cm⁻², 5 min). **f** Particle size distribution of ARNGs@TMZ/ICG with and without

NIR irradiation (808 nm, 0.5 W cm⁻², 5 min). **g** Young's modulus of ARNGs@TMZ/ICG after receiving different time of NIR irradiation (808 nm, 0.5 W cm⁻²) measured by AFM (n = 3 independent samples). Data are presented as mean ± SD (one-way ANOVA and Tukey's multiple comparison test). **h**, **i** In vitro release profile of TMZ (**h**) and ICG (**i**) from ARNGs@TMZ/ICG in pH 7.4 phosphate buffer at 37 °C, with and without NIR irradiation (808 nm, 0.5 W cm⁻²) (n = 3 independent experiments). Data are presented as mean ± SD.

Fig. 2d). The DLS measurement confirmed that the particle size of the nanogels decreased significantly after irradiation (Fig. 2e), further verifying the NIR-induced disintegration of NGs@TMZ/ICG.

The degradation of PDDA can be achieved by various photosensitizers as well as different types of ROS[40]. To examine the degradation of PDDA in nanogels, we loaded another classic photosensitizer Rose Bengal (RB) into the nanogels, and exposed them to 520 nm irradiation. With the increase of irradiation time, the color of the nanogels faded gradually, leaving only the color of RB (Fig. 2f). More interestingly, the fluorescence of RB in the nanogels was very weak at the beginning, due to the quench by PDDA through fluorescence resonance energy transfer. However, the fluorescence of RB boosted after the nanogels were exposed to the irradiation (bottom pictures, Fig. 2f), evidencing that the conjugated backbone of PDDA had been broken by the light irradiation (Supplementary Fig. 1). The chemical structural change of PDDA in NIR-induced disintegration of NGs@TMZ/ICG was further evaluated using Raman spectroscopy (Fig. 2g). The pristine NGs@TMZ/ICG exhibited two strong Raman peaks corresponding to the C=C bond (1520 cm⁻¹) and C≡C bond

(2120 cm⁻¹) of PDDA backbone. Upon NIR irradiation, these two peaks decreased gradually yet synchronously (Fig. 2h), indicating the simultaneous cleavage of C=C and C≡C bonds in PDDA backbone during the decomposition. Collectively, we have successfully fabricated PDDA-based nanogels, and the ROS generated by photosensitizers upon NIR irradiation can decompose PDDA efficiently, hence allowing for the laser-controlled disintegration of the nanogels.

**Preparation and NIR-induced deformation of ARNGs@TMZ/ICG**
The prepared NGs@TMZ/ICG was then camouflaged with the ApoE peptide-decorated erythrocyte membranes (AR) by mechanical sonication to obtain the final biomimetic nanogels (ARNGs@TMZ/ICG). These biomimetic nanogels were deformable upon NIR irradiation (Fig. 3a). The AR-coated nanogels, including ARNGs@TMZ/ICG and the single drug loaded ARNGs@TMZ and ARNGs@ICG, had an average size of around 190 nm (Fig. 3b), slightly larger than the corresponding non-coated nanogels. In addition, their surface charges became more negative, evidencing the successful camouflaging by the membranes (Fig. 3c). We next examined whether NIR irradiation could induce the

deformation of ARNGs@TMZ/ICG for the desired drug release. We studied the morphologies of ARNGs@TMZ/ICG before and after NIR irradiation using TEM imaging. The pristine nanogels were spherical with clearly visible erythrocyte membranesgraph (Fig. 3d). However, after the 808 nm laser irradiation, although their erythrocyte membranes were preserved, the particles were deformed with irregular and fused shapes (Fig. 3e). Interestingly, DLS measurement (Fig. 3f) demonstrated that the particle size of ARNGs@TMZ/ICG increased significantly after the irradiation. The Young's moduli of ARNGs@TMZ/ICG, measured using atomic force microscopy (AFM), significantly reduced after the NIR irradiation, further evidencing the light-induced deformation of these nanogels (Fig. 3g). The nanogels wrapped inside the erythrocyte membranes were disintegrated upon NIR irradiation, making the ARNGs much softer and looser than before the irradiation. Accordingly, the in vitro release of TMZ and ICG from ARNGs@TMZ/ICG both increased significantly upon 808 nm laser irradiation (Fig. 3h, 3i), evidencing that the NIR irradiation promoted the drug release from the nanogels (Supplementary Fig. 2). To further evaluate whether the NIR irradiation or the generated ROS had any influence on the chemical stability of TMZ, we quantified the TMZ before and after the light irradiation (808 nm, 0.5 W cm$^{-2}$, 5 min) using high performance liquid chromatography (HPLC). The content of TMZ in ARNGs@TMZ/ICG remained almost unchanged after the irradiation (Supplementary Fig. 3), suggesting that the NIR irradiation as well as the ROS generated by ICG did not affect the chemical stability of TMZ.

## BBB penetration, tumor uptake, and cytotoxicity

To evaluate the BBB permeability of the biomimetic nanogels, we first set up an in vitro trans-well model by seeding a monolayer of endothelial cell bEnd.3 to mimic the BBB. TMZ was not included in this study to avoid possible toxicity to endothelial cells. The trans-endothelial electrical resistance (TEER) value of the monolayer bEnd.3 remained >200 Ω cm$^2$ throughout the study, confirming the integrity of the endothelial cell monolayer[41,42]. The cumulative transport ratio of ARNGs@ICG was significantly higher than that of RNGs@ICG, the nanogels coated by erythrocyte membranes but without ApoE peptide decoration (Fig. 4a). The enhanced BBB permeability through ApoE peptide functionalization was resulted from their high binding affinity with low density lipoprotein (LDL) receptors that were overexpressed in bEnd.3 cells[43,44]. Notably, NIR irradiation did not affect the transport of ARNGs across the mimic BBB layer (Supplementary Fig. 4).

We next assessed the cellular uptake of ARNGs@ICG in U87MG human GBM cells. From confocal laser scanning microscope (CLSM) images, stronger red fluorescence was seen in cells treated with ARNGs@ICG than those treated with RNGs@ICG or free ICG, indicating their efficient cellular uptake on basis of the receptor-mediated endocytosis (Fig. 4b). Intriguingly, the ICG fluorescence in cells treated with ARNGs@ICG increased significantly after being exposed to 808 nm laser irradiation ("ARNGs@ICG + L"). Since the fluorescence of ICG was quenched inside the nanogels, the boosted ICG fluorescence unambiguously evidenced that the NIR irradiation facilitated the release of ICG from the nanogels. Flow cytometry analysis further quantified the cellular uptake of various particles (Fig. 4c). Upon NIR irradiation, the ICG fluorescence of cell treated with ARNGs@ICG was 1.5-fold higher than without NIR irradiation, and 2.8-fold higher than that of cells treated with RNGs@ICG, implying the necessity of the ApoE peptide functionalization. As a control, mixing with free ApoE peptide did not facilitate the cell uptake of the nanogels (Supplementary Fig. 5). We further evaluated the tumor uptake of ARNGs@TMZ/ICG using a 3D U87MG tumor spheroid model, and found that ICG fluorescence could be detected in the center of the tumor spheroids upon NIR irradiation. As a comparison, considerably weaker fluorescence was observed for the treatment without irradiation (Fig. 4d), indicating that NIR irradiation promoted the deep penetration of ICG to distal tumor cells. It should be noted that the

NIR-triggered ICG release from ARNGs@TMZ/ICG may also magnify the permeability of the nanogels. Interestingly, non-targeting nanogels, with or without NIR irradiation, both displayed weak ICG fluorescence, indicating that the ApoE functionalization also promoted the permeability of nanogels (Fig. 4d).

The ability of ARNGs@ICG to generate ROS upon 808 nm light irradiation was examined using U87MG cells stained with 2,7-dichlorodihydrofluorescein diacetate (DCFH-DA). As shown in Fig. 4e, when exposed to NIR irradiation, the cells treated with ARNGs@ICG manifested stronger green fluorescence than those treated with RNGs@ICG or free ICG, indicating an efficient ROS production. The CellTiter-Lumi™ luminescent cell viability assay showed that ARNGs@TMZ/ICG upon NIR irradiation (ARNGs@TMZ/ICG + L) resulted in more efficient tumor cell inhibition than the monotherapies (ARNGs@TMZ and ARNGs@ICG + L, Fig. 4f), evidencing that NIR had activated ARNGs@TMZ/ICG for a synergy of combined photodynamic and chemotherapy on GBM. The combination index between PDT and chemotherapy were below 0.5 when TMZ and ICG concentrations were from 0.125 to 80 μg mL$^{-1}$ (Supplementary Fig. 6). The propidium iodide (PI) and Annexin V co-staining analysis further confirmed that ARNGs@TMZ/ICG + L triggered cell apoptosis efficiently (Fig. 4g). Collectively, NIR irradiation effectively activated ARNGs@TMZ/ICG through enhanced BBB penetration, tumor uptake and cytotoxicity, which were essential for the subsequent in vivo alleviation of GBM.

## Pharmacokinetics, in vivo BBB penetration, and biodistribution

The pharmacokinetics of intravenously injected nanogels was examined by measuring the plasma concentrations of TMZ and ICG in tumor-free mice. The elimination half-life of TMZ (t$_{1/2}$, β) was 8.1 h and 7.4 h for ARNGs@TMZ/ICG and RNGs@TMZ/ICG respectively, markedly longer than that of NGs@TMZ/ICG (4.7 h) (Fig. 5a). It should be noted that both ARNGs@TMZ/ICG and RNGs@TMZ/ICG maintained a high TMZ concentration for a period of up to 48 h, demonstrating that camouflaging with erythrocyte membrane could remarkably increase the circulation time of nanogels, which was important for sufficient drug accumulation in the tumor sites. The pharmacokinetic profile of ICG was similar to that of TMZ (Fig. 5b). As expected, the mixed solution of TMZ and ICG (Free TMZ/ICG) were rapidly excreted from the body with a short half-life of less than 10 min, consistent with previous reports[45,46].

We next evaluated the ability of different nanogels to traverse BBB in vivo, following a single tail vein injection in the orthotopic luciferase expressed U87MG (U87MG-Luc) tumor-bearing mouse model. 808 nm laser irradiation was applied on the tumor site at 4 h post-injection, and the ICG fluorescence was monitored using an in vivo imaging system. As shown in Fig. 5c, the fluorescence of ICG quickly increased in brain within 2 h for mice treated with ARNGs@TMZ/ICG, either with or without NIR irradiation. As a biodegradable polysaccharide, the degradation of pullulan could also influence the deformation of nanogels. However, pullulan degrades slowly in physiological conditions, given the low levels of extracellular pullulanase[47]. The above results evidenced that the NIR-induced nanogel deformation was dominant in the site-specific burst release of the therapeutic cargos in this study. The ICG fluorescence co-localized very well with the tumor bioluminescence, implying excellent targeting efficiency of ARNGs@TMZ/ICG. More interestingly, the ICG fluorescence increased remarkably upon NIR irradiation (ARNGs@TMZ/ICG + L) and maintained at a high level for up to 24 h, implying that the NIR irradiation triggered the burst drug release in tumor lesions. In contrast, both non-targeting RNGs@TMZ/ICG and uncoated NGs@TMZ/ICG displayed much weaker ICG fluorescence in brain, regardless NIR irradiation, suggesting their limited BBB penetration and tumor accumulation.

The biodistribution of TMZ and ICG delivered via various nanogels was examined by ex vivo images of the major organs taken

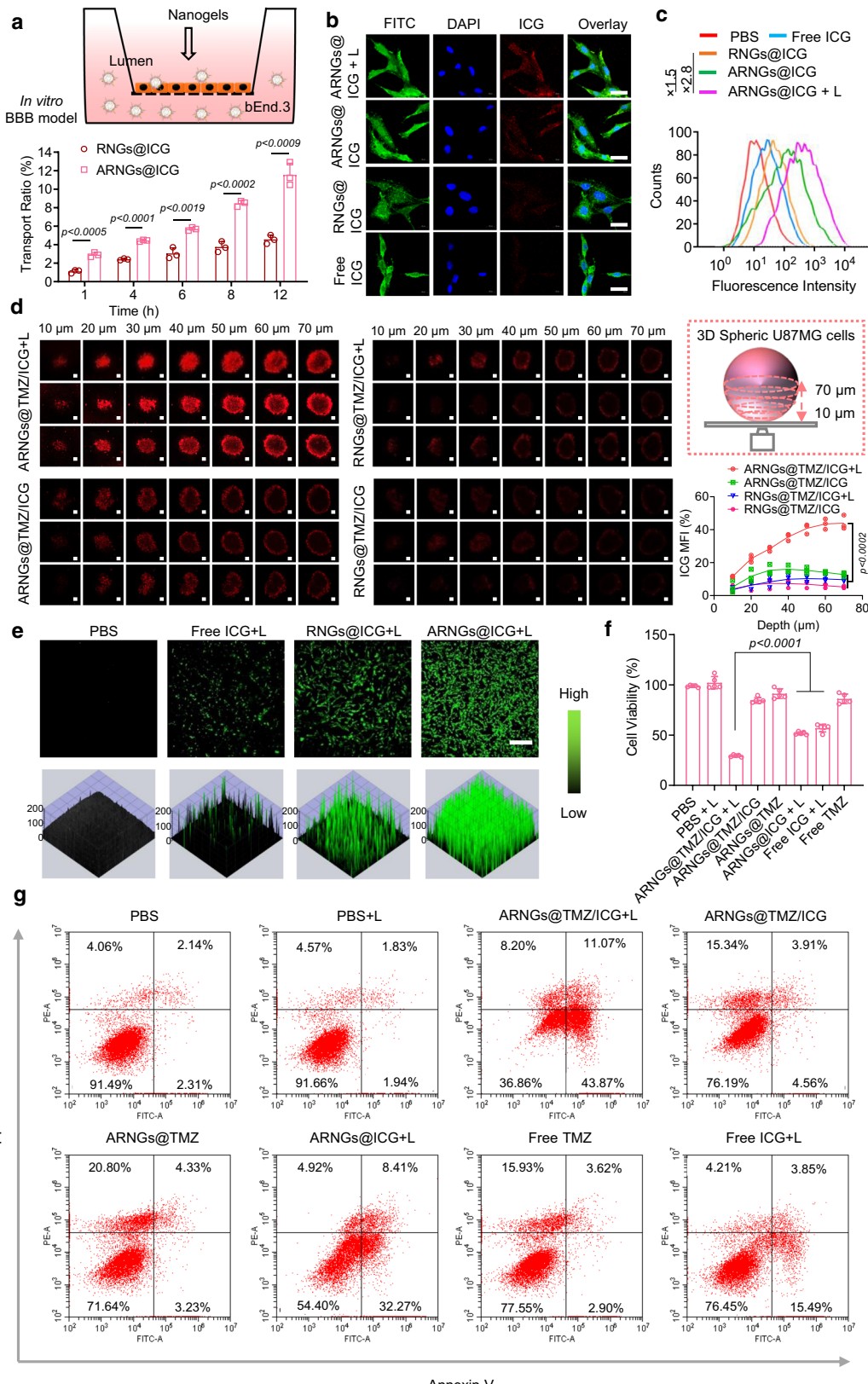

from the mice receiving different treatments (Fig. 5d). The mouse treated with ARNGs@TMZ/ICG showed obviously stronger ICG fluorescence in brain than the mice treated with other nanogels, and NIR irradiation could induce even higher ICG fluorescence (ARNGs@TMZ/ICG + L). The exceptional BBB penetration as well as tumor accumulation of ARNGs@TMZ/ICG were mainly attributed to

the receptor-mediated transcytosis, in that LDL receptor family was overexpressed by both endothelial cells of BBB and U87MG brain tumor cells (Supplementary Fig. 7), which had a strong binding affinity with ApoE peptide (Fig. 5e). In addition, the CLSM images of tumor tissue slides presented enhanced ICG fluorescence at locations that were more distal from the tumor boundary for

**Fig. 4 | In vitro BBB penetration, cellular and tumor uptake, and cytotoxicity of biomimetic nanogels. a** Cumulative transport ratio of ARNGs@ICG and RNGs@ICG across of the in vitro BBB trans-well model at different time ($n = 3$). Data are presented as mean ± SD (one-way ANOVA and Tukey's multiple comparison test). **b** CLSM images of U87MG cells receiving various treatments and stained by fluorescein 5-isothiocyanate (FITC) and 4',6-diamidino-2-phenylindole dihydrochloride (DAPI). Scale bar: 20 μm. **c** Flow cytometry analysis of U87MG cells receiving various treatments. **d** Uptake of ARNGs@TMZ/ICG by U87MG multicellular spheroids after 4 h of incubation with and without NIR irradiation. Z-stack imaging was progressed from the bottom into the core of the spheroids at an interval of 10 μm. Scale bar: 500 μm; Top right: Schematic diagram of the 3D spherical U87MG model; Bottom right: Quantification of the relative ICG mean fluorescence intensity (MFI, $n = 3$). Data are presented as mean ± SD (one-way ANOVA and Tukey's multiple comparison test). **e** CLSM images of DCFH-DA-stained U87MG cells after incubating with various nanogels for 4 h upon NIR irradiation. Scale bar: 100 μm. **f** Cell viability of U87MG cells by CellTiter-Lumi™ luminescent cell viability assay at 48 h after receiving various treatments ($n = 5$). Data are presented as mean ± SD (one-way ANOVA and Tukey's multiple comparison test). **g** Apoptosis analysis of U87MG cells by flow cytometry at 48 h after receiving various treatments and stained by PI and Annexin V. For all studies, incubation time with treatment agents: 4 h; NIR: 808 nm, 0.5 W cm⁻², 5 min; ICG concentration: 10 μg mL⁻¹; TMZ concentration: 10 μg mL⁻¹.

ARNGs@TMZ/ICG with NIR irradiation than without NIR irradiation (Fig. 5f). Interestingly, the total amounts of TMZ and ICG in brain at 8 h post treatment, combining released and unreleased drugs in the nanogels, were similar between mice treated with ARNGs@TMZ/ICG + L and ARNGs@TMZ/ICG (Fig. 5g, h). However, the ICG fluorescence in mice treated with ARNGs@TMZ/ICG boosted upon NIR irradiation (ARNGs@TMZ/ICG + L, Fig. 5c, d), implying that the enhanced ICG release from the nanogels as well as the distal tumor drug transfer occurred after the nanogels had been accumulated in tumor site.

## Complete suppression of orthotopic U87MG tumors in mice

To evaluate the anti-tumor effect of NIR-activatable ARNGs@TMZ/ICG, we administered various nanogels into orthotopic U87MG-luc bearing mice via tail vein injection on Day 10, 12, 14, 16, and 18 after the tumor implantations (Fig. 6a). NIR irradiation was applied at 4 h after each injection for the mice receiving light treatment. The treatment of orthotopic brain tumor was preliminarily evaluated based on the intensity of tumor bioluminescence. As shown in Fig. 6b, the tumor bioluminescence in the brains of mice treated with ARNGs@TMZ/ICG and NIR irradiation (ARNGs@TMZ/ICG + L) remained almost unchanged, exhibiting supreme tumor suppression by this treatment. As a comparison, the mice receiving monotherapy ARNGs@TMZ or ARNGs@ICG + L displayed less efficient anti-tumor effect, and PBS treatments could hardly restrain tumor proliferation with or without laser irradiation. The quantitative analysis of the bioluminescence intensity further verified the complete inhibition of GBM growth by NIR-activated ARNGs@TMZ/ICG (Fig. 6c), whereas the anti-tumor activity of ARNGs@TMZ/ICG was significantly compromised without NIR activation. Notably, there was no obvious body weight loss for mice treated with NIR-activated ARNGs@TMZ/ICG, demonstrating that the treatment had few side effects to mice (Fig. 6d). In contrast, dramatic body weight reduction was observed for mice treated with PBS, mainly due to the tumor-associated brain dysfunctions. Of note, the median survival time of the mice treated with NIR-activated ARNGs@TMZ/ICG was 69 days, which was remarkably longer than those of the mice treated with inactivated ARNGs@TMZ/ICG (43 days), ARNGs@TMZ (34 days), or ARNGs@ICG with NIR irradiation (38 days) (Fig. 6e). The histological analysis of the whole brain using hematoxylin-eosin (H&E) staining showed that NIR-activated ARNGs@TMZ/ICG resulted in minimized tumor size in brain, consistent with the lowest tumor bioluminescence intensity by this treatment (Fig. 6f). Terminal deoxynucleotidyl transferase dUTP nick-end labeling (TUNEL) analysis (Fig. 6g) showed that NIR-activated ARNGs@TMZ/ICG induced the highest levels of apoptosis (caspase 3) and nucleus damage (γH2AX) in tumor cells, and the cell proliferation signal (Ki67) was also the weakest among all the groups (Supplementary Fig. 8). Moreover, NIR-activated ARNGs@TMZ/ICG treatment showed the strongest ROS generation, which also contributed to the excellent anti-tumor efficacy of the biomimetic nanogels (Supplementary Fig. 9). Additionally, NIR-activated ARNGs@TMZ/ICG exhibited negligible side effects to the major normal tissues including heart, liver, spleen, lung, and kidney (Supplementary

Fig. 10), demonstrating their good biocompatibility. To further evaluate if there was any brain damage induced by ICG, we stained the astrocyte and microglia of mice brains after the study. All mice treated with the ICG-containing nanogels, either with or without NIR irradiation, displayed similar astrocyte and microglia morphologies as those receiving PBS treatment (Supplementary Fig. 11), suggesting that the ICG-containing nanogels caused negligible adverse side effects on normal brains.

## Efficient inhibition of orthotopic CSC2 glioblastoma stem cells (GSCs) tumors in mice

The excellent anti-tumor effects of NIR-activated nanogels encouraged us to further investigate whether they could also restrain the incurable GSCs. Firstly, we established a mice model bearing orthotopic patient-derived GSCs CSC2 tumors that expressed luciferase stably (Fig. 7a, b). The overexpression of LDL receptors of CSC2 cells was verified to lay the fundamental of the efficient BBB crossing as well as the active internalization of our biomimetic nanogels (Supplementary Fig. 7). We next studied the cytotoxicity of NIR-activated ARNGs@TMZ/ICG on CSC2 cells, which efficiently inhibited the cell proliferation with a considerably low cell viability (Supplementary Fig. 12). The tumor bioluminescence images showed that the NIR-activated ARNGs@TMZ/ICG could effectively suppress the tumor growth during the treatment period. ARNGs@TMZ/ICG without irradiation also inhibited the tumor proliferation to some extent, but was less efficient than that with light treatment (Fig. 7b). In sharp contrast, rapid tumor growth was observed for mice treated with PBS, with or without light irradiation. The quantitative bioluminescence intensity results were in line with the images and further confirmed the best anti-tumor efficacy of NIR-activation nanogels (Fig. 7c). The H&E staining of whole brain further indicated that the mice treated with NIR-activated ARNGs@TMZ/ICG had the smallest tumor size (Fig. 7d). Furthermore, the mice in the experiment all maintained their body weight throughout the study (Fig. 7e). Importantly, NIR-activated ARNGs@TMZ/ICG markedly prolonged the mice survival with a median survival time of 63 d, which was significantly longer than the mice receiving PBS (20 d) or nanogels without NIR (44 d, Fig. 7f). The histological analysis of tumor slices showed that the NIR-activated nanogels caused the most tumor cell apoptosis and the least tumor cell proliferation (Fig. 7g). Additionally, major organs in mice treated with NIR-activated ARNGs@TMZ/ICG showed no side effects (Supplementary Fig. 13). Collectively, the NIR-activated ARNGs@TMZ/ICG exhibited an anti-tumor effect in GSC CSC2 mice model as good as in U87MG mice models, demonstrating the excellent efficacy of our NIR-activatable biomimetic nanogels.

## Biocompatibility evaluation of ARNGs@TMZ/ICG

We further assessed the biosafety of the NIR-activatable biomimetic nanogels using blood routine and blood biochemistry analysis. ARNGs@TMZ/ICG were intravenously administrated into healthy mice, and the blood was collected and detected on Day 0, 2, 4, 7, and 14 post-injection, respectively. ARNGs@TMZ/ICG showed negligible impact on the blood parameters including white blood cell (WBC), red blood cell (RBC), hemoglobin (HGB), alkaline

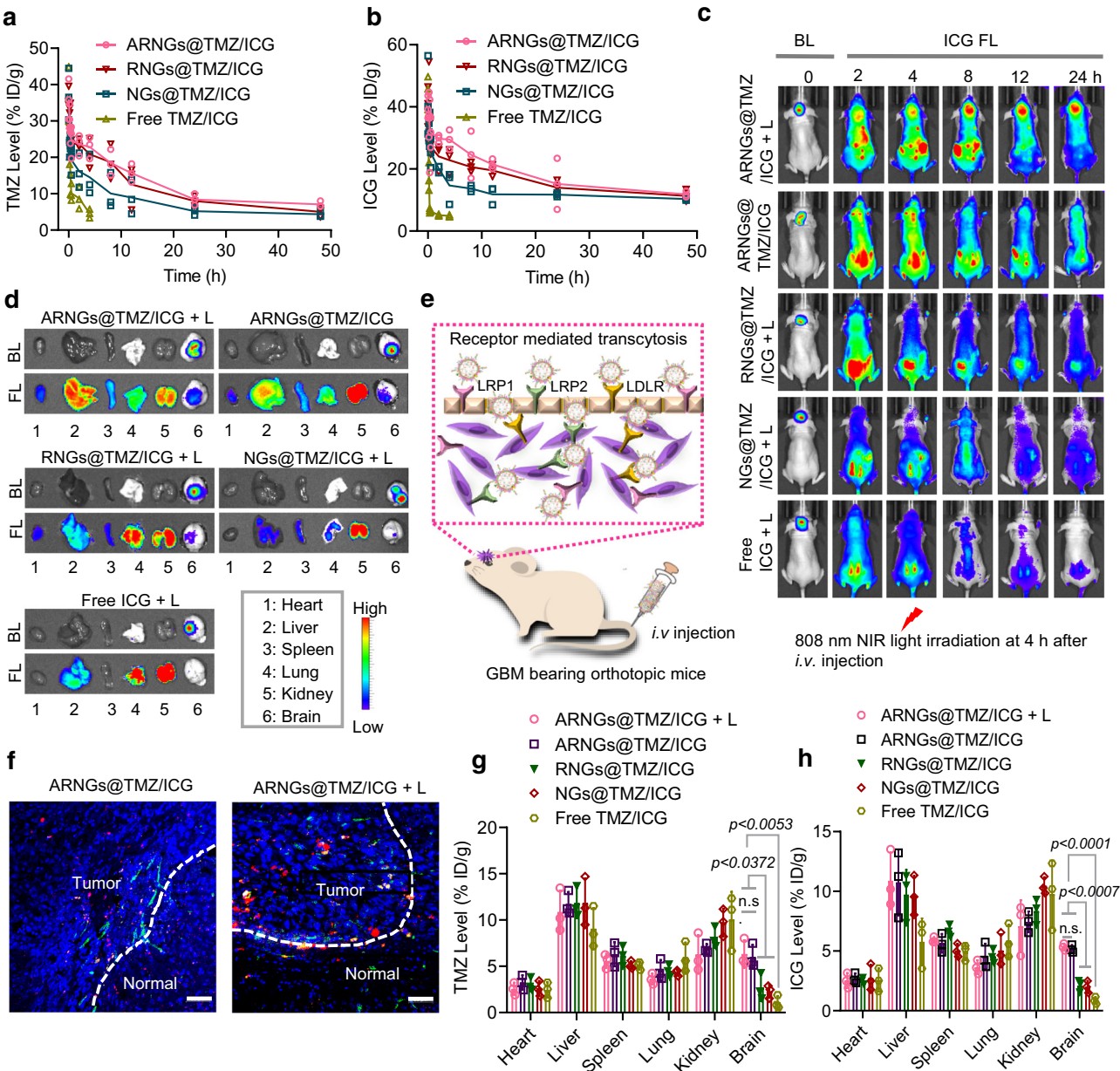

**Fig. 5 | Pharmacokinetics, in vivo BBB penetration, and biodistribution of NIR-activatable ARNGs@TMZ/ICG. a, b** Pharmacokinetics of TMZ (**a**) and ICG (**b**) from ARNGs@TMZ/ICG in healthy BALB/c mice, TMZ and ICG levels were determined by HPLC and UV-vis absorption spectra, respectively, and expressed as injected dose per gram (%ID g$^{-1}$) ($n = 3$ biologically independent samples). Data are presented as mean ± SD. **c** The IVIS images of the tumor-bearing mice showing tumor biolumi-nescence (BL) and ICG FL at different time after receiving various treatments. **d** ICG FL images of the major organs (heart, liver, spleen, lung, kidney, and brain) taken from the tumor bearing mice at 8 h after receiving various treatments shown in **c**. **e** Schematic illustration of the uptake of ARNGs@TMZ/ICG by specifically targeting to LDL receptors overexpressed on both BBB endothelial cells and brain tumor cells. **f** Fluorescence images of tumor tissue slides from the mice treated with ARNGs@TMZ/ICG and ARNGs@TMZ/ICG + L. The tumor slices were taken from the mice at 8 h post injection. The green fluorescence referred to CD31-labeled tumor blood vessels. Scale bar: 50 μm. **g, h** Quantification of TMZ (**g**) and ICG (**h**) accu-mulation in different organs from the tumor bearing mice at 8 h after receiving different treatments ($n = 3$ biologically independent samples). Data are presented as mean ± SD (one-way ANOVA and Tukey multiple comparisons tests). In studies of Fig. 3c–h, orthotopic U87MG-Luc tumor bearing mice were used, and all agents were intravenously administered. The dosage was 10 mg kg$^{-1}$ for both TMZ and ICG. NIR irradiation (808 nm, 1 W cm$^{-2}$, 5 min) was applied at 4 h post-injection.

phosphatase (ALP), alanine aminotransferase (ALT) and aspartate aminotransferase (AST) (Fig. 8a–f). The blood urea, creatinine (CREA) and carbonic anhydrase (UA) level also exhibit no significant change during the 2 weeks after ARNGs@TMZ/ICG administration (Supplementary Fig. 14). Pro-inflammatory cytokines such as IL-1β, IL-6, and TNF-α were assessed in liver and kidney (Fig. 8g–l), which demonstrated no significant difference between PBS, Free TMZ/ICG, and ARNGs@TMZ/ICG treatment groups after two weeks. However, the Free TMZ/ICG group showed significantly increased pro-inflammatory cytokines on Day 2. Collectively, the biocompat-ibility evaluation pointed out that encapsulating TMZ and ICG in our NIR-activatable biomimetic nanogels could effectively reduce the systemic adverse effects of the drugs.

## Discussion

To maintain an effective drug concentration in deep tumor lesions remains an unsurmountable challenge in GBM treatment. In the efforts described above, we have developed a NIR-activatable

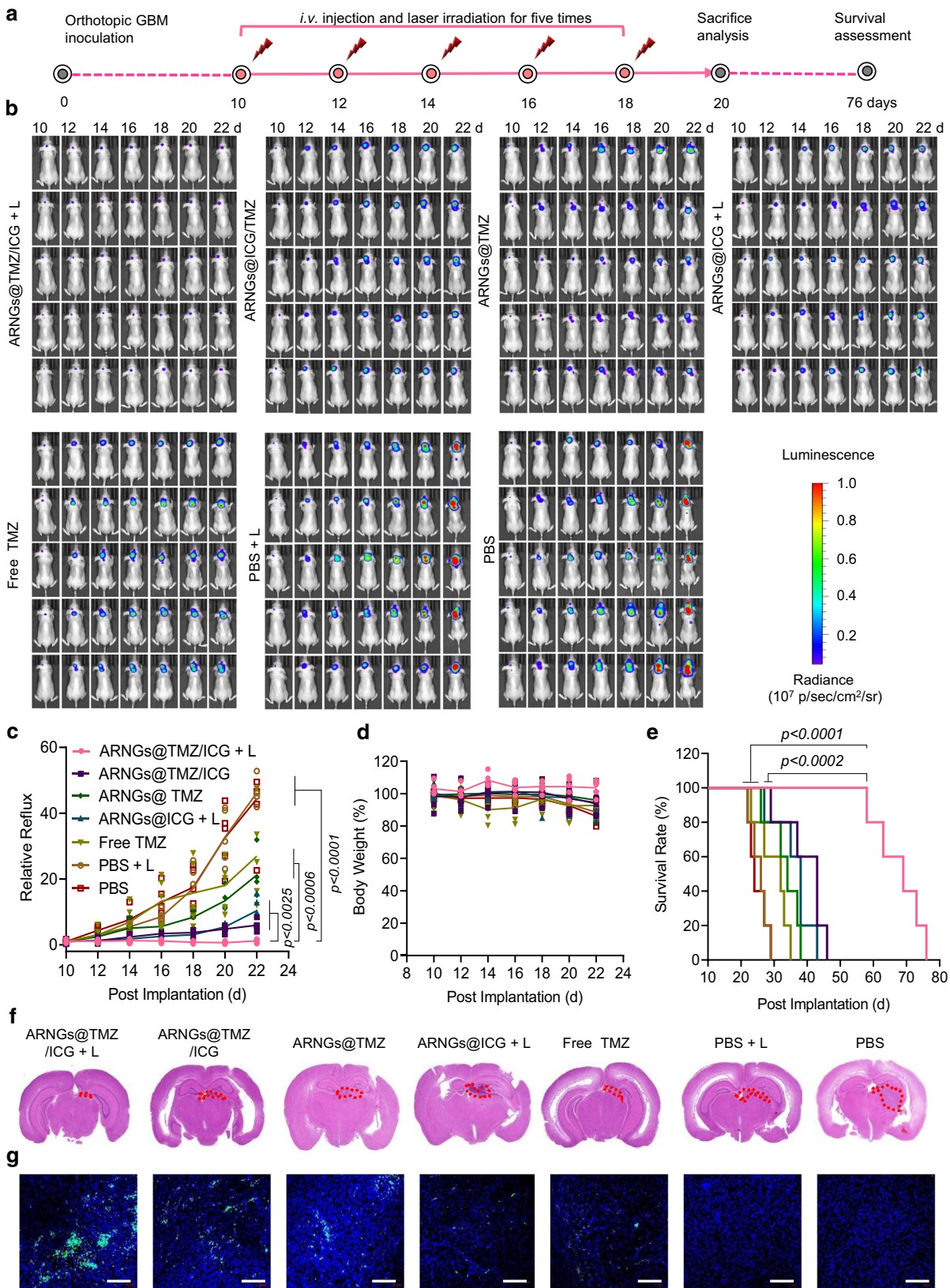

biomimetic nanogel system ARNGs@TMZ/ICG that realizes the deep penetration of TMZ and ICG to distal GBM tumor cells across BBB and other biological barriers. In our design, we highlight the precise activation of ARNGs@TMZ/ICG by ICG-generated ROS upon NIR irradiation. Compared with other ROS-responsive drug delivery systems, the PDDA-based nanogels are inert to

endogenous oxidative conditions, which is important to enhance their stability in the circulation system and avoid possible drug leakage in non-tumor microenvironments. The decoration with ApoE-peptide functionalized erythrocyte membrane further extends the circulation time, improves tumor accumulation, and facilitates the BBB penetration of the nanogels. After the nanogels

**Fig. 6 | Anti-tumor efficacy of NIR-activatable ARNGs@TMZ/ICG against orthotopic GBM. a** Schematic illustration of the timeline of the anti-tumor efficacy studies on orthotopic GBM bearing mice. U87MG-Luc cells were orthotopically inoculated into the brains of 6–8 weeks nude mice. On Day 10 after the tumor inoculation, mice with a similar bioluminescence intensity were selected and randomized into 7 groups ($n = 5$). Various formulations were intravenously injected at a dose of 10 mg TMZ equiv. kg⁻¹ and 10 mg ICG equiv. kg⁻¹ on Day 10, 12, 14, 16, and 18 post-tumor implantation. **b** In vivo bioluminescence images of orthotopic GBM in live mice receiving different treatments. **c** Quantified tumor bioluminescence levels of orthotopic GBM in each group ($n = 5$ biologically independent samples). Data are presented as mean ± SD (one-way ANOVA and Tukey multiple comparisons tests). **d** Body-weight changes in mice ($n = 5$ biologically independent samples). Data are presented as mean ± SD. **e** Kaplan–Meier analysis of the mice ($n = 5$ biologically independent samples). One-way ANOVA and Tukey multiple comparisons tests. **f** H&E-staining images of the orthotopic brain tumor tissues excised from the mice in each group. **g** TUNEL-staining images of the orthotopic brain tumor tissues excised from the mice in each group. Scale bars: 100 μm.

reach effective tumor accumulation, which can be traced by ICG fluorescence, NIR irradiation is manually applied on tumor lesions. ICG then generates ROS to deform the nanogels, triggering burst localized drug release and deep penetration of the drugs to distal tumor cells. Therefore, this activation process is especially favorable to maintain a high concentration of TMZ and ICG in deep GBM lesions. Consequently, the focal synergistic NIR PDT and chemotherapy resulted in complete orthotopic GBM and GBM stem cells tumor inhibition and significantly improved survival rate, with excellent biocompatibility and minimal side effects. The NIR activation design brings in deep GBM penetration and spatiotemporally controlled drug release, endowing the biomimetic nanogels efficient anti-GBM efficacy in both orthotopic U87MG and GBM stem cells (CSC2) mice models, with nearly three-fold improvement of median survival time. Therefore, compared with interiorly responsive nanosystems decorated by similar targeting ligands[48,49], NIR activation makes our ARNGs@TMZ/ICG more effective in GBM inhibition. The PDDA-based NIR-activatable nanogels not only offer translational advantages as a potential therapeutic platform against malignant GBM, but also pave a way to engineer precisely controllable therapies to treat a variety of diseases based on active response towards physiological signals.

## Methods
### Ethical regulations
All animal handling protocols and experiments were approved by the Medical and Scientific Research Ethics Committee of Henan University School of Medicine (P. R. China) (HUSOM-2018-355).

### Materials
PDDA was synthesized following a previous reported procedure[39]. Pullulan was purchased from Meihua Group (Hebei, China). mPEG$_{2k}$-OH was purchased from Sigma-Aldrich LLC. N-(3-Dimethylaminopropyl)-N′-ethylcarbodiimide Hydrochloride (EDC) was purchased from Heowns Chemical (Tianjin, China). 4-Dimethylaminopyridine (DMAP) and dimethyl sulfoxide (DMSO) were purchased from Sinopharm Chemical Reagent Co., Ltd. Indocyanine green (ICG) and Temozolomide (TMZ) were purchased from Energy Chemical. Lipid-tethered polyethylene glycol-maleimide (DSPE-PEG$_{2k}$-Mal) was purchased from Jenkem Technology (Beijing, China). Apolipoprotein E peptide [ApoE-SH, (LRKLRKRLL)2C, 95%] was purchased from China peptide Co., Ltd. (Suzhou, China). BCA protein assay kit (Catalog no. P0012), ROS measurement DCFH-DA kit (Catalog no. S0033S), One step TUNEL apoptosis assay kit (Catalog no. C1086), and Annexin V-FITC and PI apoptosis detection kit (Catalog no. C1062S) were purchased from Beyotime Biotechnology Co., Ltd. Antibodies used: Western blotting: LDL Receptor Rabbit Monoclonal Antibody (Beyotime Biotechnology. Catalog no. AF1438, 1/1000 dilution); Rabbit monoclonal to Anti-LRP1 antibody (Abcam, Catalog no. ab92544, 1/20000 dilution); LRP2/Megalin Rabbit Polyclonal Antibody (Beyotime Biotechnology., Catalog no. AF7395, 1/500 dilution); and GAPDH Mouse Monoclonal Antibody (Beyotime Biotechnology., Catalog no. AF2819, 1/1000 dilution). Immunohistochemistry staining: Ki67 rabbit polyclonal antibody (Servicebio, Catalog no. GB111499, 1/500 dilution); Cleaved-Caspase-3 rabbit polyclonal antibody (Servicebio, Catalog no. GB11532, 1/500

dilution); and γH2A.X rabbit polyclonal antibody (Affinity, Catalog no.AF6187, 1/500 dilution). Immunofluorescence analysis: Anti-Iba1 rabbit monoclonal antibody (Abcam, Catalog no. ab178846, use a concentration of 0.2 μg mL⁻¹) and GFAP rabbit polyclonal antibody (Protecntech, Catalog no.16825-1-AP, 1/200 dilution).

### Cell lines
The bEnd.3 cell line and U87MG cell line were purchased from the American Type Culture Collection (ATCC). U87MG-Luc cell line was purchased from Shanghai Model Organisms Center, Inc. The CSC2 cancer stem cells (CSC2) were provided by Prof. Jong Bae Park of Specific Organs Cancer Branch, Research Institute, National Cancer Center, Goyang, Gyeonggi, Korea. The cell lines were morphologically confirmed according to the information provided by ATCC and Shanghai Model Organisms Center, Inc. The U87MG cell line and the CSC2 cells were authenticated by short tandem repeat (STR) analysis. All cell lines were maintained according to ATCC instructions. All cells were repeatedly screened for mycoplasma and maintained in culture for <6 months after receipt. Mycoplasma was negative in the cell culture medium model (Mycoplasma Detection Kit (PCR) purchased from Servicebio, Cat.G1900).

### Animals
Female BALB/c mice (6–8 weeks, 18–20 g) and female ICR mice (6–8 weeks) were purchased from SPF (Beijing) Biotechnology Co., Ltd. All animals were bred in a pathogen-free facility with a 12 h light/dark cycle at 20 ± 3 °C and 40–50% humidity and had ad libitum access to food and water. All animal handling protocols and experiments were approved by the Medical and Scientific Research Ethics Committee of Henan University School of Medicine (PR China) (HUSOM-2018-355).

### Synthesis of nanogels and loading drugs
In a typical nanogel preparation, PDDA (10 mg), pullulan (100 mg), mPEG$_{2k}$-OH (200 mg) were dissolved in 5 mL of DMSO separately. DMAP (30 mg) and EDC (150 mg) were dissolved in 15 mL of DMSO. The above solutions were mixed together (total volume 30 mL) and stirred at room temperature. The color of the reaction mixture gradually changed from light yellow to wine red. The reaction mixture was stirred for 12 h and monitored by dynamic light scattering (DLS). After the crosslinking was completed, ICG and TMZ were added to the reaction mixture to reach a concentration of 20 μg mL⁻¹ for ICG and 100 mg mL⁻¹ for TMZ. The mixture was further incubated for 3 h, and quenched with 100 mL of deionized H$_2$O before being dialyzed for 72 h with 7,000 Da molecular weight cut off (MWCO). The nanogel dispersion was then diluted to desired concentrations for future studies.

### Preparation of apolipoprotein E (ApoE) peptide-decorated erythrocyte membrane
Whole blood withdrawn via eye socket bleeding of female ICR mice (6–8 weeks) was added with EDTA (1.5 mg mL⁻¹) for anticoagulation and centrifuged for removal of plasma and buffy coat. The sediment red blood cells (RBCs) were washed 3-4 times by ice cold 1 × PBS, and suspended in hypotonic medium of ice cold 0.25 × PBS for 30 min. The

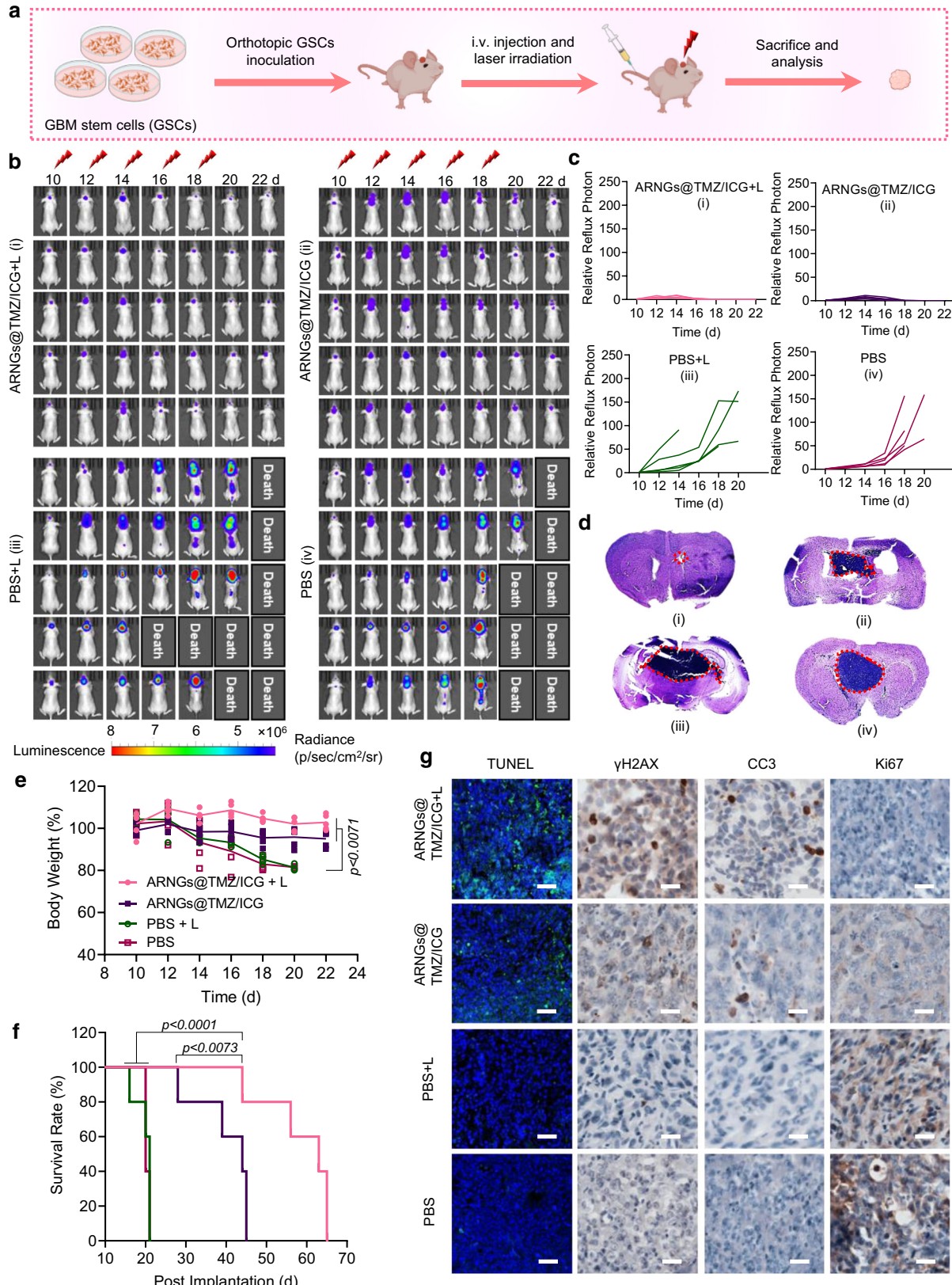

released hemoglobin was discarded by centrifuging the solution at 11,000 g for 5 min. The RBC membranes were collected after being washed 3-4 times by 1 × PBS. The as prepared RBC membranes were resuspended in 1 × PBS and sonicated for 5 min, and the vesicles were extruded serially through 400 nm and then 200 nm polycarbonate porous membranes using an Avanti mini extruder (Avanti Polar Lipids). On the other hand, ApoE and DSPE-PEG-Mal at a molar ratio of 1.5:1 were dissolved in 1 × PBS (pH 7.4) and reacted at 37 °C for 24 h. The produced DSPE-PEG-ApoE was isolated through dialysis (MWCO 7,000 Da) against deionized $H_2O$ for 48 h to remove free ApoE and

**Fig. 7 | Anti-tumor efficacy of NIR-activatable ARNGs@TMZ/ICG against orthotopic GBM stem cells (GSCs) mice. a** Schematic illustration of the establishment of GSCs mice model. CSC2-Luc cells were orthotopically inoculated into the brains of 6–8 weeks nude mice. On Day 10 after the tumor inoculation, mice with a similar bioluminescence intensity were selected and randomized into 4 groups (n = 5). Various formulations were intravenously injected at a dose of 10 mg TMZ equiv. kg⁻¹ and 10 mg ICG equiv. kg⁻¹ on Day 10, 12, 14, 16, and 18 post tumor implantation. **b** In vivo bioluminescence images of orthotopic GSCs in live mice receiving different treatments. **c** Quantified tumor bioluminescence levels of orthotopic GSCs in each group. **d** H&E-staining images of the orthotopic brain tumor tissues excised from the mice in each group. **e** Body-weight changes in mice (n = 5 biologically independent samples). Data are presented as mean ± SD (one-way ANOVA and Tukey multiple comparisons tests). **f** Kaplan–Meier analysis of the mice (n = 5 biologically independent samples). One-way ANOVA and Tukey multiple comparisons tests. **g** TUNEL, γH2AX, CC3, and Ki67 staining images of the orthotopic brain tumor tissues excised from the mice in each group. Scale bar: 200 μm for TUNEL image and 60 μm for γH2AX, CC3, and Ki67 images.

followed by lyophilization. The degree of ApoE conjugation was determined to be 98% by Micro BCA protein assay kit (Thermo scientific). To form ApoE peptide-decorated erythrocyte membrane, 100 μL of the extruded vesicles were incubated with 50 μg DSPE-PEG-ApoE at 37 °C for 30 min.

### Preparation of ARNGs@TMZ/ICG nanogels

To encapsulate NGs@TMZ/ICG into ApoE-decorated erythrocyte membrane, 200 μL of obtained ApoE-decorated erythrocyte membrane were mixed with NGs@TMZ/ICG (1 mg) in 1 mL of deionized H₂O. The pH value of the mixture was adjusted to 7.4 with NaOH solution. The mixture was then votexed and extruded through a 200 nm polycarbonate porous membrane for 7 times with an Avanti mini-extruder. The as prepared ARNGs@TMZ/ICG nanogels were then dialyzed against water to remove free DSPE-PEG-ApoE.

### Elasticity measurement by AFM

AFM force measurements were conducted using a MultiMode 8 AFM (Bruker) with an inverted optical microscope. The nanogels in 1 × PBS dispersion with or without light irradiation were dropped onto a monocrystalline silicon substrate. MLCT probe (Bruker) and cantilever A were used ($f_0 = 22$ kHz, $k = 0.07$ N m⁻¹). The force curves were measured in shooting mode, with the Z piezo working in a closed loop. Young's modulus of the nanogels was calculated by applying the Sneddon-modified Hertz model.

### Drug release in vitro

1 mL of ARNGs@TMZ/ICG was loaded into dialysis tubing (MWCO 12,000–14,000), which was merged into 10 mL of 1 × PBS (pH 7.4) as dissolution media (TMZ: 10 μg mL⁻¹, ICG: 10 μg mL⁻¹). The mixture was then incubated at 37 °C under continuous shaking. At predetermined time points of 1 h, 2 h, 4 h and 6 h, the corresponding mixtures were treated with or without laser irradiation (808 nm, 0.5 W cm⁻²) for 5 min. The dialysate was taken out to analyze the amount of drug released, while the same amount of fresh 1 × PBS was added back and kept shaking for further study. The ICG concentration in the samples was measured by UV/vis spectroscopy and the TMZ levels were determined by HPLC (Agilent 1200) with UV detection at 329 nm using a mixture of methanol/0.5% acetic acid (v/v) as eluent (v/v = 10/90). The stability of ARNGs@TMZ/ICG under NIR irradiation (808 nm, 0.5 W cm⁻², 5 min) was assessed by the HPLC under the same condition as described.

### BBB penetration studies in vitro

A 12-well transwell plate with 0.4 μm of mean pore size membrane was used to establish the in vitro BBB model. The bEnd.3 cells (1 × 10⁴ cells per well) were seeded in the transwell insert with a diameter of 12 mm. The transendothelial electrical resistance (TEER) was detected by a Millicell-ERS volt-ohmmeter to monitor the cell monolayer integrity during the cell culture process. A TEER value above 200 Ω cm² was suitable for further experiments. ARNGs@ICG and RNGs@ICG (ICG: 10 μg mL⁻¹) were added to the cells, respectively. The medium in both apical and basolateral chambers was collected at 1 h, 4 h, 6 h, 8 h, and 12 h to measure the ICG fluorescence.

### Flow cytometry assays

U87MG cells were evenly seeded into 6-well plates (1 × 10⁵ cells per well) and incubated over night at 37 °C. Various formulations were added next (as indicated in the manuscript) and incubated for another 4 h. NIR irradiation was applied in the corresponding groups (808 nm, 0.5 W cm⁻²) for 5 min. The cells were incubated for another 2 h and rinsed with 1 × PBS for three times immediately before being analyzed by flow cytometry (Accuri, BD, USA). The competitive cell uptake was evaluated by pretreating the cells with free ApoE peptide (200 μg mL⁻¹) for 2 h.

### Cellular uptake

U87MG cells were seeded on the cover-slide system at a density of 5 × 10⁴ cells per well and cultured overnight for cell attachment. Afterward, the cells were treated with ARNGs@ICG, RNGs@ICG, and free ICG (ICG: 10 μg mL⁻¹) and incubated for 4 h. The ARNGs@ICG group was irradiated with the laser (808 nm, 0.5 W cm⁻²) for 5 min and incubated for another 2 h. The cells were washed three times with 1 × PBS and fixed with cold 4% paraformaldehyde for 0.5 h. The cells were stained with 5 μg mL⁻¹ FITC-glycine cyclopeptide at room temperature for 30 min, and then with 1 μg mL⁻¹ DAPI for 10 min. CLSM was applied to observe cellular internalization.

### Penetration evaluation in 3D-spheroid tumor model

The 3D tumor spheroids of U87MG were established according to the following steps. U87MG cells were maintained in the standard culture conditions. After being treated with 0.25% trypsin, the cells were detached and suspended in DMEM (10% FBS) and the cell density was adjusted to 10⁴ cells mL⁻¹. Added into the prime surface 96UZ plate (MS-9096UZ). After the cells were cultured for two days, the cells began to form balls. ARNGs@TMZ/ICG and RNGs@TMZ/ICG (TMZ: 10 μg mL⁻¹, ICG: 10 μg mL⁻¹) were added to the culture medium and then treated with or without light irradiation (808 nm, 0.5 W cm⁻², 5 min) after 4 h. The mixtures were incubated for another 2 h, and the tumor spheroids were washed with PBS about 3 times. The permeability of different nanoparticles into tumor spheroids was studied by CLSM (Zeiss 980, 10 × magnification).

### Reactive oxygen species (ROS) generation

For in vitro evaluation, 1 × 10⁵ of U87MG cells were seeded the confocal plates. The cells were co-incubated with ARNGs@ICG, RNGs@ICG, and free ICG (ICG: 10 μg mL⁻¹) with or without light irradiation (808 nm, 0.5 W cm⁻², 5 min) after 4 h. The cells were then incubated with 10 μM DCFH-DA for 30 min before being washed by 1 × PBS. ROS was measured by a commercial assay kit (Catalog no. S0033S) under CLSM (Zeiss 880). The in vivo ROS detection was similar as the above, except using the brain tumor slices taken from mice treated with different formulations.

### Cytotoxicity studies

U87MG or CSC2 cells were inoculated into a 96-wells plate (1 × 10⁴ per well) with cells attaching to the wall for 24 h. ARNGs@TMZ/ICG, ARNGs@TMZ, ARNGs@ICG, free TMZ, free ICG (TMZ: 10 μg mL⁻¹, ICG: 10 μg mL⁻¹) and blank ARNGs were then incubated with U87MG cells, respectively. after 4 h of incubation, NIR laser (808 nm, 0.5 W cm⁻²) was applied on corresponding cells for 5 min, and the cells were

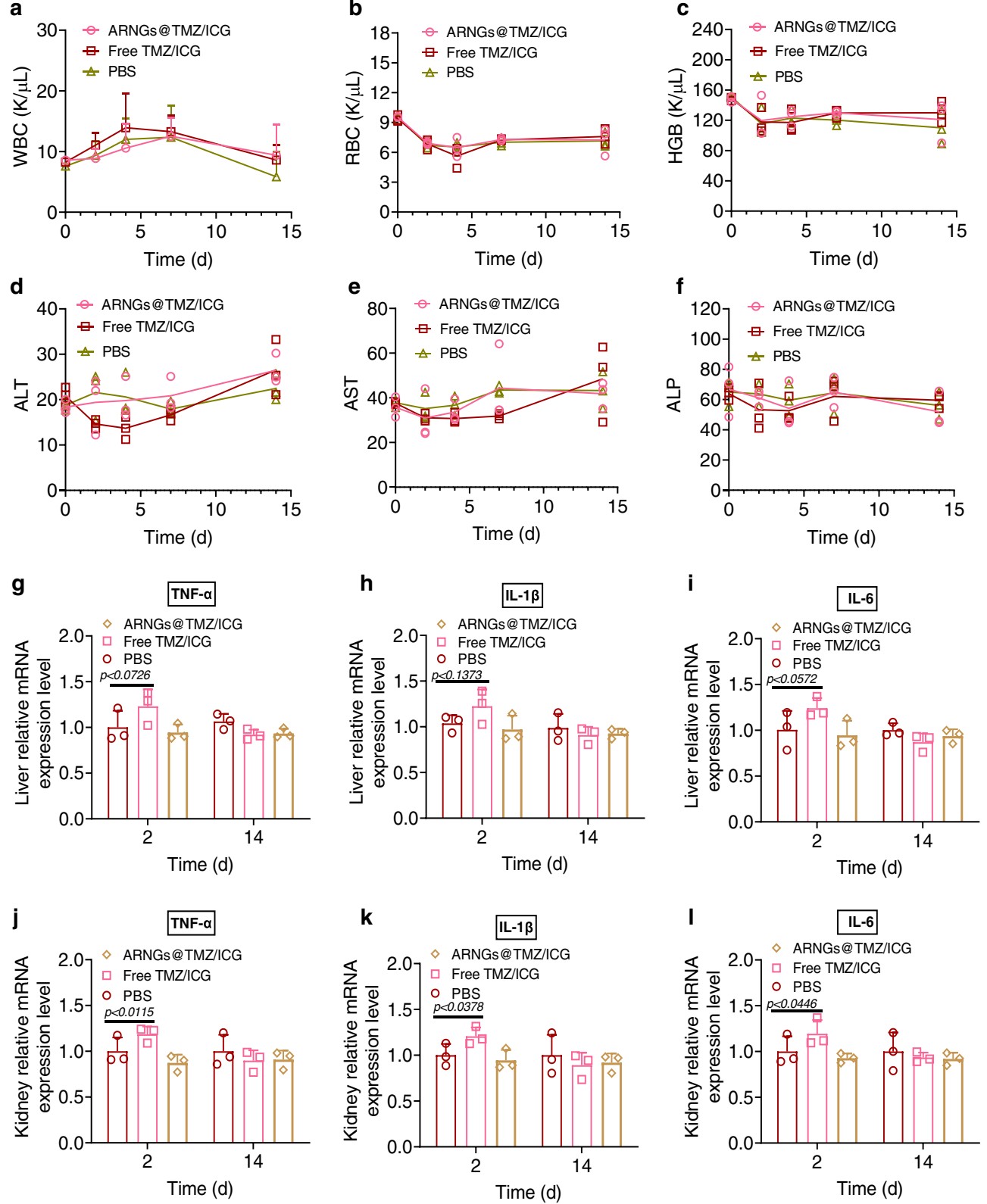

**Fig. 8 | Biocompatibility evaluation of NIR-activatable ARNGs@TMZ/ICG.**
**a**–**c** Cytotoxicity and in vivo biocompatibility assessment of ARNGs@TMZ/ICG. White blood cell (WBC) (**a**), red blood cell (RBC) (**b**) and hemoglobin (HGB) (**c**) levels in blood samples were assessed after a single dose tail vein injection. Data are presented as mean ± SD. **d**–**f** Blood chemistry examinations. Plasma alkaline phosphatase (ALP) (**d**), alanine aminotransferase (ALT) (**e**) and aspartate aminotransferase (AST) (**f**) levels were assessed after a single dose tail vein injection. Data are presented as mean ± SD. **g**–**l** Expression of sentinel proinflammatory cytokines TNF-α, Il-1β and Il-6 in liver (**g**–**i**) and kidney (**j**–**l**) assessed on Day 2 and 14 after a single dose tail vein injection. Data are presented as mean ± SD (*n* = 4 independent samples, one-way ANOVA and Tukey multiple comparisons tests).

incubated for another 44 h. Viability of the cells was analyzed using the CellTiter-Lumi™ luminescent cell viability assay kit.

The inhibitory concentration (IC$_{50}$) of ARNGs@TMZ/ICG was carried out in U87MG cells similarly using a series drug concentration (0.125, 2.5, 5, 10, 20, 40 and 80 µg mL$^{-1}$). CI values were calculated using the Chou-Talalay method with CompuSyn software.

## Evaluation of apoptosis by flow cytometry
U87MG cells were seeded in 24-well plates at a density of $5 \times 10^4$ cells per well. After overnight, the cells were treated with ARNGs@TMZ/ICG, RNGs@TMZ/ICG and free TMZ/ICG (TMZ: 10 µg mL$^{-1}$, ICG: 10 µg mL$^{-1}$) for 4 h. Then the culture medium was refreshed, and the cells were incubated with fresh DMEM medium at 37 °C with or without laser irradiation (808 nm, 0.5 W cm$^{-2}$) for 5 min and incubated for another 44 h. The cells receiving no treatment were tested as the negative control. All the cells were washed with 1 × PBS for three times, digested by trypsin (EDTA depleted), and collected by centrifugation. After being washed with 1 × PBS for three times, the cells were resuspended in 0.5 mL of annexin binding buffer. After that, all cells were stained by the binding buffer containing PI and Annexin-V-FITC for 15 min and finally measured by flow cytometry (BD FACSAria TM III). The gating strategy was shown in detail in Supplementary Fig. 15.

## Pharmacokinetic evaluation in vivo
BALB/c female tumor-free mice at 6–8 weeks were randomly divided into four groups ($n = 3$). The mice in each group were individually injected with ARNGs@TMZ/ICG, RNGs@TMZ/ICG, NGs@TMZ/ICG, and free TMZ/ICG. The dosages were 10 mg TMZ equiv. kg$^{-1}$ and 10 mg ICG equiv. kg$^{-1}$. Blood was taken via eye socket bleeding with capillary tubes at different time points about 0 min, 3 min, 10 min, 30 min, 1 h, 2 h, 4 h, 8 h, 12 h, 24 h and 48 h. The amount of TMZ and ICG was determined by HPLC and fluorescence spectroscopy, respectively.

## Brain tumor modeling
An orthotopic U87MG glioblastoma bearing mouse model was established with a high success rate of nearly 100% via implantation of minced glioblastoma tissue into the left striatum of BALB/c nude mice. Briefly, U87MG-Luc cells ($1 \times 10^5$ cells) 5 µL were implanted into the left striatum (2 mm lateral to the bregma and 3 mm deep) of anesthetized animals using a 24# trocar and a specifically made propeller. Then the burr hole was filled with bone wax (Johnson & Johnson International, Brussels, Belgium) and the scalp was closed with tissue glue (3 M Animal Care Products, St Paul, Minnesota, USA). The growth of the glioma was monitored by bioluminescence using an imaging system (IVIS, Lumina III; Caliper, MA, USA), at 10 min after the mice were anesthetized and injected with luciferase substrate D-luciferin potassium (15 mg mL$^{-1}$ in 1 × PBS) at 75 mg kg$^{-1}$. The mouse was euthanatized when its brain tumor exceeded the pre-specified maximal tumor volume of 1 cm$^3$, or their maximal weight loss exceeded 20%, which had been strictly complied with the animal ethics. The anesthetic mice were euthanatized by carbon dioxide.

Luciferase expressed CSC2 GBM stem cells (CSC2-Luc) mice model was established following our previous procedure[50]. Similar as described above, except using CSC2-Luc cells instead of tiny tumor tissues.

## Fluorescence imaging in vivo
ARNGs@TMZ/ICG, RNGs@TMZ/ICG, free TMZ/ICG in 200 µL of 1 × PBS were intravenously injected into the orthotopic U87MG-Luc glioblastoma tumor-bearing mice via tail veins at 0, 2, 4, 8, 12 and 24 h post injection, and then monitored using a NIR fluorescence imaging system ($\lambda_{ex} = 780$ nm; $\lambda_{em} = 831$ nm). The NIR irradiation was applied at 4 h post injection using an 808 nm laser (1 W cm$^{-2}$, 5 min). The dosages were 10 mg TMZ equiv. kg$^{-1}$ and 10 mg ICG equiv. kg$^{-1}$.

## Biodistribution of ARNGs@TMZ/ICG nanogels
ARNGs@TMZ/ICG, RNGs@TMZ/ICG, and free TMZ/ICG were intravenously injected into the mice bearing orthotopic U87MG-Luc GBM via tail veins. The dosages were 10 mg TMZ equiv. kg$^{-1}$ and 10 mg ICG equiv. kg$^{-1}$. The mice were applied laser irradiation (808 nm, 1 W cm$^{-2}$, 5 min) at 4 h post-injection and sacrificed at 8 h post-injection. The major organs including heart, liver, spleen, lung, kidney and cancerous brain were collected and washed in 1 × PBS. Fluorescence images were acquired by the IVIS Lumina III system. To quantify the amounts of TMZ and ICG being delivered to different organs, the organs were homogenized in 0.6 mL of 1% triton X-100 with a homogenizer at 70 Hz for 10 min. Each tissue lysate was incubated with 0.9 mL DMSO at room temperature overnight. The TMZ and ICG in the supernatant was determined by HPLC (as described above) and fluorescence spectroscopy based on a calibration curve, respectively and expressed as injected dose per gram of tissue (% ID g$^{-1}$).

## Tumor penetration
Two groups of mice bearing orthotopic U87MG-Luc human glioblastoma tumor were injected with ARNGs@TMZ/ICG via the tail vein, and treated with or without NIR laser (808 nm, 1 W cm$^{-2}$, 5 min) at 4 h post-injection. The tumor slices were taken from the mice at 8 h post injection and observed by CLSM. The green fluorescence labeled CD31 antibody was used to label tumor blood vessel to observe the depth of penetration of the nanogels into the tumor.

## Tumor inhibition experiment
Female BALB/c nude mice at age of 6–8 weeks were selected and inoculated with orthotopic GBM tumors. To visualize U87MG-Luc or CSC2-Luc cells, luciferin was injected intraperitoneally (75 mg kg$^{-1}$) at 10–15 min before imaging. The mice were weighed and randomly divided into nine groups (n = 8, 5 for monitoring survival and 3 for histological analysis): ARNGs@TMZ/ICG with or without laser, RNGs@TMZ/ICG with laser, ARNGs@TMZ, ARNGs@ICG with laser, free TMZ, free ICG with laser, PBS with or without laser. The dosages were 10 mg TMZ equiv. kg$^{-1}$ and 10 mg ICG equiv. kg$^{-1}$. The NIR irradiation was performed at 4 h after injection (808 nm, 1 W cm$^{-2}$, 5 min). The different formulations were administered on day 10, 12, 14, 16, and 18 after tumor inoculation. The tumor changes of the mice were observed with the IVIS system every two days, and the body weight was weighed. The survival rate of the mice in each group was determined using the Kaplan-Meier method. On Day 20, the brain, liver, heart, spleen, lung, and kidney were collected from one mouse in each treatment group for histological analysis by H&E staining. Tumor tissues were also stained with terminal deoxynucleotidyl transferase dUTP nick-end labeling (TUNEL) and observed under CLSM (Zeiss 880). Immunohistochemistry (IHC) analysis was performed by labeling with cleaved caspase 3 (CC3), γH2AX and proliferation (Ki67) antibodies and visualized using a digital microscope (Leica). The staining was conducted following the manufacturer's instructions (Servicebio, Wuhan). Immunofluorescence analysis of microglia and astrocyte were stained with Iba1 and GFAP antibodies (both were FITC), respectively. DAPI was used for nuclear staining. IBA1 antibody (abcam, ab178846), GFAP antibody (protecntech, 16825-1-AP).

## Blood biochemistry and blood routine examinations
Normal BALB/c female mice at the age of 6–8 weeks were randomly divided into three treatment groups ($n = 3$). PBS, free TMZ/ICG and ARNGs@TMZ/ICG (The dosages were 10 mg TMZ equiv. kg$^{-1}$ and 10 mg ICG equiv. kg$^{-1}$) were intravenously injected into mice via the tail vein. At prescribed time points after injection, blood was collected via eye socket bleeding. For blood biochemistry examination, whole blood was centrifuged at 800 g for 5 min to collect serum for analysis. Blood biochemistry and routine blood test were conducted by Servicebio Technology Co., Ltd. (Wuhan, China) with an automated

chemistry analyzer (Chemray 240 Rayto Inc.). Blood cell parameters were analyzed with an automated blood cell analyzer (BC-2800Vet-Mindray Inc.).

## Statistics and reproducibility

All the analysis data are given as mean ± SD. The results were analyzed by the Student's *t* test between two groups. Exact p values were provided accordingly in the figures. For the data in Figs. 2d, 3d, e, 6g, 7g and Supplementary Figs. 7, 9, 10, 11, 13. three experiments were repeated independently with similar results and results from representative experiments were shown.

## Reporting summary

Further information on research design is available in the Nature Portfolio Reporting Summary linked to this article.

## Data availability

All data generated or analyzed during this study are included in this published article and its Supplementary Information file and the Source Data file. Source data are provided as a Source Data file.

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

## Acknowledgements

This work was supported by the National Natural Science Foundation of China (NSFC 51803049, 21877042, U2004171, 31800841, 22107032, and 32071388), the National Key Technologies R&D Program of China (2018YFA0209800 and 2018YFA0208903), Program for Science & Technology Innovation Talents in Universities of Henan Province (21HASTIT033). We specially thank Professor Jong Bae Park of the Specific Organs Cancer Branch, Research Institute, National Cancer Center of Korea for providing us the CSC2 cells.

## Author contributions

D.Z. and S.T. contributed equally to this work. Y.Z., B.S., and L.L. designed the experiments, supervised the project, and revised the manuscript. D.Z., S.T., and Y.L. prepared and characterized the nanogels and performed in vitro and in vivo experiments. D.Z., S.T., and Y.L. analyzed the data. D.Z. and S.T. wrote the manuscript and revised it with comments from M.Z, X.Y., Y.Z., B.S., and L.L. All authors participated in discussions throughout the project.

## Competing interests

The authors declare no competing interests.
