## [Peer Review File · Nature Communications]

Title: Near Infrared-Activatable Biomimetic Nanogels
Enabling Deep Tumor Drug Penetration Inhibit
Orthotopic GlioblastomaREVIEWER COMMENTS

Reviewer #1 (expertise in nanotherapy, nanoparticle coating, APOE, glioblastoma) (Remarks to the Author):

In this manuscript, the authors reported an interesting and innovative NIR-activatable biomimetic nanogel delivery system for effective GBM treatment. One of the building blocks of the nanogel PDDA can selectively degrade with light-induced reactive oxygen species (ROS). These nanogels showed prolonged circulation time, enhanced deep tumor accumulation in GBM lesions, and NIR- controlled activation. Importantly, the nanogels enabled synergistic combinations of photodynamic and chemotherapy, which completely inhibited the growth of orthotopic GBM mouse model with extended survival time. These biomimetic nanogels provide an interesting therapeutic paradigm for the treatment of GBM. The study is rather comprehensive. I would recommend considering this manuscript after addressing the following concerns:

1. The degradation of PDDA in the presence of NIR-irradiated ICG is interesting. Is it possible that the degradation could be achieved by other photosensitizers? Or other sources of ROS?
2. Besides PDDA, pullulan is also a biodegradable polysaccharide. The author should discuss the in vivo degradation of pullulan and its influence on the deformation of the nanogels.
3. The nanogel can consume the ROS generated by ICG, so it is important to give some evidence that the ICG could generate excess ROS to degrade the nanogel and achieve PDT at the same time.
4. ICG is a photosensitizer with both anti-cancer photodynamic and photothermal effects. The authors need to comment on the photothermal effect of ICG in this study.
5. The biomimetic nanogels demonstrate stronger fluorescence in 3D U87MG spheroids and mice models as compared with non-irradiation counterparts, what's the performance of non-targeting nanogels? Moreover, the deeper penetration of these nanogels received light may also partly ascribe to the accelerated release of ICG, which should also be clarified.
6. The biomimetic nanogels inhibited orthotopic U87MG xenografts efficiently as evidenced by the levels of tumor cell apoptosis (TUNEL, Caspase 3, H2AX) and proliferation (Ki-67), while the ROS generation was lacking.
7. Will the ROS generated by ICG affect the chemical stability of TMZ? The authors need to check the stability of TMZ when the nanogel is irradiated by NIR.
8. There are a couple of reports on apoE/angiopoep-2-mediated nano-delivery of chemotherapeutics for the treatment of GBM in which the nanoparticles are responding to the reductive environment in the GBM tumor cells (Int. J. Nanomed. 2021, 16, 4105-4115; J. Control. Release 2018, 278, 1-8). It would be interesting to have appropriate comparison and discussion about what are the features or advantages of the system reported here.
9. A figure exemplifying the gating strategy of flow cytometry analysis should be provided.

Reviewer #2 (expertise in glioblastoma, nanoparticles, ICG) (Remarks to the Author):

- Q1. The target design ApoE has been done a lot and there is no innovation in the field of brain targeting.
- Q2. Whether the photosensitizer ICG has toxic and side effects on the brain and how to evaluate it.
- Q3. Does the design of the delivery system have any special advantages for tumor cell uptake?
- Q4. In Figure 3 d, e, after near-infrared light irradiation, why does the red blood cell membrane outside the hydrogel fall off, and what is the principle?

Reviewer #3 (expertise in glioblastoma and temozolomide treatment) (Remarks to the Author):

This is an interesting article evaluating a novel nanoparticle formulation that releases drug cargo in response to NIR irradiation. Inclusion of an ApoE peptide targeting uptake across the BBB, this strategy is a promising and interesting strategy. Overall, the studies are clearly presented and appear carefully performed. However, there are several limitations that dampen enthusiasm for the manuscript that can be readily addressed.

Sole use of U87 is a limitation to the study. Confirming key studies in a second glioma cell line would be useful. Notably, patient-derived xenograft or glioma stem cell models are more current and applicable strategies.

Figure 4F/G. Incubation with TMZ for 24h is insufficient to evaluate cytotoxicity specifically from TMZ, which occurs over 2-3 cell cycles. The concentration of TMZ in the various conditions should be specified.

Tetrazolium salt-based assays, such as MTT, XTT MTS assays, are affected by the mitochondrial effects of TMZ and are not accurate assessment of TMZ toxicity. A orthogonal assay should be used to confirm the results.

Cytotoxicity is associated with ICG alone and TMZ alone. Synergy analyses could be useful to define the impact of the combined nanoparticle simply reflects additive toxicities.

The ApoE decoration for the nanoparticles could be quite important for BBB and tumor targeting. In this context, creating and testing nanoparticles with a scrambled peptide sequence but the same amino acid content is an important control that should be used in key experiments.

Throughout the in vitro studies, the effective concentration of TMZ should be provided for both free and nanoparticle formulated drug treatments.

Figure 7 – the changes in ALT and AST are lower in the free TMZ treatment as compared to placebo or nano formulation. These changes in ALT and AST are subtle and unlikely to be clinically meaningful.

Point-by-Point Responses to Reviewers' Comments

We would like to thank the editor and reviewers for their positive and constructive comments, which are greatly helpful to improve the quality of our work. We have completed a number of additional experiments in the past months and carefully revised our manuscript according to the reviewers' comments. Point-by-point responses are as below:

Reviewer #1

General comments: In this manuscript, the authors reported an interesting and innovative NIR-activatable biomimetic nanogel delivery system for effective GBM treatment. One of the building blocks of the nanogel PDDA can selectively degrade with light-induced reactive oxygen species (ROS). These nanogels showed prolonged circulation time, enhanced deep tumor accumulation in GBM lesions, and NIR-controlled activation. Importantly, the nanogels enabled synergistic combinations of photodynamic and chemotherapy, which completely inhibited the growth of orthotopic GBM mouse model with extended survival time. These biomimetic nanogels provide an interesting therapeutic paradigm for the treatment of GBM. The study is rather comprehensive. I would recommend considering this manuscript after addressing the following concerns.

1. The degradation of PDDA in the presence of NIR-irradiated ICG is interesting. Is it possible that the degradation could be achieved by other photosensitizers? Or other sources of ROS?

Response: We really appreciate the reviewer for the elaborate review and helpful comments. The degradation of PDDA can be achieved by other photosensitizers than ICG as well as different sources of ROS. In our previous work on PDDA degradation (*J. Am. Chem. Soc.* **2021**, *143*, 10054-10058), we have used three representative conditions to initiate PDDA degradation, including Methylene Blue (MB) with 640 nm light irradiation, Rose Bengal (RB) with 520 nm light irradiation, and Fe³⁺-mediated Fenton reaction. Both MB and RB could generate singlet oxygen upon light irradiation, while the Fenton reaction could typically generate hydroxy radicals. In the presence of all these types of ROS, we observed complete degradation of PDDA with succinic acid as one of the major degradation products (Fig. R1). Accordingly, we have added one sentence in the revised manuscript on Page 6 (highlight in red): **The degradation of PDDA can be achieved by various photosensitizers as well as different types of ROS.**

Fig. R1 a, b ^1H NMR spectra (a) and ^{13}C NMR spectra (b) of pristine PDDA and crude PDDA degradation mixture under three different degradation conditions (1. Fenton reaction, 2. MB with irradiation, 3. RB with irradiation, SA: succinic acid). c HR-MS of crude PDDA degradation mixture under the corresponding degradation conditions.

2. Besides PDDA, pullulan is also a biodegradable polysaccharide. The author should discuss the in vivo degradation of pullulan and its influence on the deformation of the nanogels.

Response: We highly appreciate the reviewer's nice comment. It is true that pullulan can also degrade in certain biological conditions. In particular, pullulan can degrade in the presence of pullulanase, and the degradation rate depends on the molecular weight of pullulan and the activity of pullulanase (*Enzyme Res.* **2012**, 2012, 921362). However, the degradation of pullulan is very slow in normal physiological environments such as blood circulation, given the low level of pullulanase in the extracellular environment (*J. Bioact. Compat. Polym.*, **1995**, 10, 299-312; *Mater. Sci. Eng., C*, **2016**, 58, 1046-1057). After being crosslinked in a nanogel, the biodegradation of pullulan is further inhibited, and the cell membrane coating can also improve the stability of the nanogel. Therefore, in the first few hours after injection, before we applied 808 nm laser to trigger the nanogel deformation, the biodegradation of pullulan should be restrained during this period. In our study, after receiving NIR irradiation, the nanogels exhibited significantly enhanced TMZ and ICG release rates, which further evidenced that the photo-induced nanogel deformation was dominant in the site-specific burst release of the therapeutic cargos in this study. We have added this discussion in the revised manuscript on Page 15 (highlighted in red): As a biodegradable polysaccharide, the degradation of pullulan could also influence the deformation of nanogels. However, pullulan degrades slowly in

physiological conditions, given the low levels of extracellular pullulanase. The above results evidenced that the NIR-induced nanogel deformation was dominant in the site-specific burst release of the therapeutic cargos in this study.

3. The nanogel can consume the ROS generated by ICG, so it is important to give some evidence that the ICG could generate excess ROS to degrade the nanogel and achieve PDT at the same time.

Response: We thank the reviewer for the very helpful suggestion. As the reviewer suggested, we used 9,10-anthracenediyl-bis(methylene) dimalonic acid (ABDA) as a probe to detect the excess ROS generation by ICG after being loaded in the nanogels (Fig. R2). We observed relatively slower degradation of ABDA by ICG loaded in nanogels than by free ICG, which was consistent with the consumption of ROS by PDDA in the nanogels. Moreover, the degradation of ABDA by ICG-loaded nanogels undoubtedly evidenced that excess ROS could be generated by the ICG within PDDA-pullulan nanogels. In addition, once ICG was released from the nanogels, its generated ROS could be no longer consumed by PDDA in the nanogels, so that these ROS could efficiently kill tumor cells during PDT. In our manuscript, Fig. 4f demonstrated that “ARNGs@ICG+L” resulted in tumor cell inhibition, which also proved the cytotoxicity of ROS generated by ICG in nanogels.

Fig. R2 a, b Singlet oxygen generation of ICG in PDDA-pullulan nanogels (a) and free ICG (b) as detected by ABDA. **c** Quantitative comparison of the absorption peak of ABDA at 380 nm. (ABDA concentration is 50 μM , ICG concentration is 10 $\mu\text{g mL}^{-1}$).

4. ICG is a photosensitizer with both anti-cancer photodynamic and photothermal effects. The authors need to comment on the photothermal effect of ICG in this study.

Response: We appreciate the reviewer’s nice comment. As pointed out by the reviewer, the therapeutic effect of ICG is a combination of both PDT and PTT (*Lasers Surg. Med.*, **2003**, 33,

296-310; *Nanoscale*, **2019**, 11, 6384-6393; *ACS Nano*, **2013**, 7, 2056-2067). For in vivo photo therapy, it is very difficult to completely distinguish the tumor inhibitory effects of these two simultaneous therapeutic effects. Our in vitro cellular experiment at room temperature (Fig. 4f) suggested that the PDT effect of the nanogels alone could trigger significant cell inhibition. In addition, we also evaluated the photothermal effect of ICG at the maximal brain concentration ($10 \mu\text{g mL}^{-1}$, Fig. R3), as estimated based on the drug biodistribution results (Fig. 5h). The photothermal effect of ICG at this concentration only elevated the temperature of the solution for 2°C compared with the PBS control. Therefore, the photothermal effect of ICG should be minor in our study. Moreover, considering the ROS-responsive deformation of NGs, the PDT effect of ICG not only kills tumor cells directly, but also trigger the controlled release of the loaded ICG and TMZ. Collectively, the role of the PDT effects was dominant in this spatiotemporally controlled GBM treatment.

Fig. R3 Photothermal effect of ICG at $10 \mu\text{g mL}^{-1}$.

5. The biomimetic nanogels demonstrate stronger fluorescence in 3D U87MG spheroids and mice models as compared with non-irradiation counterparts, what's the performance of non-targeting nanogels? Moreover, the deeper penetration of these nanogels received light may also partly ascribe to the accelerated release of ICG, which should also be clarified.

Response: We really thank the reviewer for the constructive comments. As the reviewer suggested, we have evaluated the tumor penetration of the biomimetic nanoparticles, including non-targeting nanogels, in 3D tumor spheroids. The results showed that biomimetic nanogels NIR-activated ARNGs@TMZ/ICG had the deepest penetration of $70 \mu\text{m}$. In sharp contrast, non-targeting nanogels ARNGs@TMZ/ICG, with or without light irradiation, demonstrated much weaker ICG fluorescence (Fig. R4, Fig. 4d in the revised manuscript), indicating their poor permeability. All these data supported that both ApoE ligand functionalization and light

irradiation are beneficial to the deep penetration of the nanogels. Accordingly, we have added the discussion in the revised manuscript on Page 12 (highlighted in red): Interestingly, non-targeting nanogels, with or without NIR irradiation, both displayed weak ICG fluorescence, indicating that the ApoE functionalization also promoted the permeability of nanogels (Fig. 4d). In addition, we agreed with the reviewer that the light triggered ICG release may facilitate the penetration of nanogels. To clarify it, we have added more discussion on Page 12 (highlighted in red): It should be noted that the NIR-triggered ICG release from ARNGs@TMZ/ICG may also magnify the permeability of the nanogels.

Fig. R4 Uptake of ARNGs@TMZ/ICG by U87MG multicellular spheroids after 4 h of incubation with and without NIR irradiation. Z-stack imaging was progressed from the bottom into the core of the spheroids at an interval of 10 μm . Scale bar: 500 μm ; bottom left: Schematic diagram of the 3D spherical U87MG model; bottom right: Quantification of the relative ICG mean fluorescence intensity (MFI, n = 3).

6. The biomimetic nanogels inhibited orthotopic U87MG xenografts efficiently as evidenced by the levels of tumor cell apoptosis (TUNEL, Caspase 3, H2AX) and proliferation (Ki-67),

while the ROS generation was lacking.

Response: We thank the reviewer for the helpful comment. As suggested by the reviewer, we have evaluated the ROS generation of tumor after receiving various treatments, using the ROS assay kit. The images observed by CLSM showed that the tumor tissue treated by either ARNGs@TMZ/ICG+L or ARNGs@ICG+L exhibited stronger DCF fluorescence than by other treatments, including ARNGs@TMZ/ICG and ARNGs@TMZ (Fig. R5, Supplementary Fig. 9 in the Supplementary Information). The results were in line with their anti-tumor effects and also supported that the efficient tumor inhibition of NIR-activated ARNGs@TMZ/ICG was partly attributed to excessive ROS generated upon NIR activation. In accordance, we have included these results in the revised manuscript on Page 19 (highlighted in red): **Moreover, NIR-activated ARNGs@TMZ/ICG treatment showed the strongest ROS generation, which also contributed to the excellent anti-tumor efficacy of the biomimetic nanogels (Supplementary Fig. 9).**

Fig. R5 CLSM images of Tumor slices excised from orthotopic U87MG-Luc human glioblastoma tumor-bearing nude mice following different treatments. Scale bar: 100 μ m.

7. Will the ROS generated by ICG affect the chemical stability of TMZ? The authors need to check the stability of TMZ when the nanogel is irradiated by NIR.

Response: We appreciate the reviewer's nice comment. To study if the chemical structure of

TMZ is affected by the NIR irradiation, we used high performance liquid chromatography (HPLC) to quantify the content of TMZ in ARNGs@TMZ/ICG before and after light irradiation. The curves showed that the content of ICG almost remained unchanged after being treated with NIR activation (808 nm, 0.5 W cm⁻², 5 min) (Fig. R6, or Supplementary Fig. 3 in the revised manuscript). Therefore, we deduced that the NIR irradiation as well as the ROS generation had negligible influence on the chemical stability of ICG. To clarify it, we have added these results in the revised manuscript on Page 9: To further evaluate whether the NIR irradiation or the generated ROS had any influence on the chemical stability of TMZ, we quantified the TMZ before and after the light irradiation (808 nm, 0.5 W cm⁻², 5 min) using high performance liquid chromatography (HPLC). The content of TMZ in ARNGs@TMZ/ICG almost remained unchanged after the irradiation (Supplementary Fig. 3), suggesting that the NIR irradiation as well as the ROS generated by ICG did not affect the chemical stability of TMZ.

Fig. R6 The high-performance liquid chromatography (HPLC) of the TMZ in ARNGs@TMZ/ICG nanogels before and after light irradiation (808 nm, 0.5 W cm⁻², 5 min).

8. There are a couple of reports on apoE/angiopoep-2-mediated nano-delivery of chemotherapeutics for the treatment of GBM in which the nanoparticles are responding to the

reductive environment in the GBM tumor cells (*Int. J. Nanomed.* 2021, 16, 4105-4115; *J. Control. Release* 2018, 278, 1-8). It would be interesting to have appropriate comparison and discussion about what are the features or advantages of the system reported here.

Response: We really appreciate the reviewer for the very inspiring suggestion. As the reviewer mentioned, there are several elaborately designs using similar targeting ligands (ApoE, angiopep-2, etc.) functionalized delivery strategies, which have achieved significantly improved blood brain barrier (BBB) penetration, brain tumor accumulation, and anti-GBM effects with prolonged survival time (*Int. J. Nanomed.* **2021**, 16, 4105-4115; *J. Control. Release* **2018**, 278, 1-8). In our work, we also adopted ApoE to decorate on the surface of the biomimetic nanogels for enhanced BBB permeability. However, we emphasized the precise NIR activation of the developed nanogels after they had accumulated in tumor lesions. The spatiotemporal control of ROS deformed the nanogels, triggered the burst localized drug release for deep penetration of the drugs to distal tumor cells. Notably, the elaborate activation process is favorable to maintain a high concentration of both TMZ and ROS in deep GBM lesions, leading to the extended survival in orthotopic U87MG mice models (69 d versus 24 d for saline), which were more effective than the previously reported nano-delivery systems based on interior glutathione-stimulated ApoE (44 d versus 23 d, *Int. J. Nanomed.* **2021**, 16, 4105-4115) or angiopep-2 (53 d versus 24 d, *J. Control. Release* **2018**, 278, 1-8). Accordingly, we have more discussion in the revised manuscript on Page 27 (highlighted in red): **The NIR activation design brings in deep GBM penetration and spatiotemporally controlled drug release, endowing the biomimetic nanogels efficient anti-GBM efficacy in both orthotopic U87MG and CSC2 GBM stem cells (GSCs) mice models, with nearly three-fold improvement of median survival time. Therefore, compared with interiorly responsive nanosystems decorated by similar targeting ligands, NIR activation makes our ARNGs@TMZ/ICG more effective in GBM inhibition.**

9. A figure exemplifying the gating strategy of flow cytometry analysis should be provided.

Response: Thank the reviewer for the valuable comment. As suggested, we have provided the detailed gating strategies for the cell uptake and apoptosis analysis of flow cytometry in Supplementary Fig. 15a and 15b, respectively (Fig. R7 below).

Fig. R7 The gating strategies of flow cytometry analysis for the cell uptake (a) and apoptosis (b) in U87MG cells.

Reviewer #2

1. The target design ApoE has been done a lot and there is no innovation in the field of brain targeting.

Response: We highly appreciate the reviewer for the elaborate review and valuable comments. We totally agree with the reviewer that ApoE functionalization strategy has been applied widely in the field of brain targeting, which indicates that the ApoE decoration is reliable in enhancing the blood brain barrier (BBB) permeability and the active tumor cell targeting capability of nanoparticles. In this manuscript, we focused on the development of a NIR-activatable biomimetic nanogels to realize the deep penetration of TMZ and ICG to distal GBM tumor cells. We emphasized the elaborate design of the nanogels to achieve spatiotemporal control of the drug release for deep tumor penetration, which was the main novelty of this work. Therefore, we needed to decorate the nanogels by a well-established and reliable targeting

ligand to warrant their efficient BBB crossing, so that the loaded TMZ and ICG could be released inside the glioma tissue in the brain as desired for distal tumor cell uptake. In this situation, we employed AopE rather than a new targeting ligand in this study, because the new one may introduce more variates and interfere with the evaluation of the drug delivery efficiency of the nanogels.

In addition to brain targeting, the insufficient drug accumulation in deep tumor sites is also one of the main challenges for the clinical treatment of GBM and other cancers (*Nat. Biomed. Eng.* 2021, 5, 481; *Nano Today*, 2021, 36, 101038). Our NIR-activatable biomimetic nanogels represents a potential solution to the problem. To better illustrate the innovation of our manuscript, we have summarized the main highlights of this work as below:

- (1) **High Permeability**: The developed biomimetic nanogels could be readily transformed under the NIR-activation with the generation of ROS in tumor lesions, further allowing the deep tumor permeability and effective drug concentration in the distal tumor cells, which directly resulted in complete GBM tumor inhibition in both orthotopic U87MG GBM and CSC2 GBM stem cell mice models.
- (2) **Low immunogenicity**: The erythrocyte membrane on the surface of nanogels endowing their low immunogenicity, which significantly improved the blood circulation time with an elimination half-live of 7.6 h.
- (3) **Good stability**: The PDDA-based nanogels inner core are inert to endogenous oxidative conditions, which exhibited better stability in circulation and lower undesired drug release as compared with many other ROS-responsive drug delivery systems to date.
- (4) **Specificity and safety**: The targeted nanogels are specifically internalized by tumor cells rather than the normal brain cells, attributed to the overexpressed receptors in GBM cells. Together with the manually controlled NIR irradiation, our nanogels could ensure negligible damage to normal organs and cells.

As a result, our developed biomimetic nanogels demonstrated excellent anti-tumor efficacy in a GBM stem cell mice model with a significant prolonged median survival time (63 days versus 21 days for PBS). We hope our explanation could better illustrate the innovation of this work and dispel the reviewer's concern.

2. Whether the photosensitizer ICG has toxic and side effects on the brain and how to evaluate it.

Response: We appreciate the reviewer's nice comment. We chose ICG in our study in that it

has been already approved by the Food and Drug Administration (FDA) for medical use. ICG has been widely used in clinical settings for diagnosis purposes with good biosafety, so that it is believed to cause very mild toxic and side effects on the brain. To further evaluate the adverse effects of ICG, we assessed the astrocyte and microglia of the brain slices following successive injections of ARNGs@TMZ/ICG nanogels. The images showed that all mice treated with the ICG-containing nanogels, either with or without NIR irradiation, displayed similar astrocyte and microglia morphologies as those receiving PBS treatment (Fig. R8, or Supplementary Fig. 11 in the Supplementary Information), which unambiguously demonstrate that ICG had negligible side effects on normal brains. To clarify it, we have included these results in the revised manuscript on Page 19 (highlighted in red): **To further evaluate if there was any brain damage induced by ICG, we stained the astrocyte and microglia of brain slices after the study. All mice treated with the ICG-containing nanogels, either with or without NIR irradiation, displayed similar astrocyte and microglia morphologies as those receiving PBS treatment (Supplementary Fig. 11), suggesting that the ICG-containing nanogels caused negligible adverse side effects on normal brains.**

Fig. R8 CLSM images of tumor slices excised from orthotopic U87MG-Luc human glioblastoma tumor-bearing nude mice following different treatments. Scale bar: 100 μ m.

3. Does the design of the delivery system have any special advantages for tumor cell uptake?

Response: We thank the reviewer for raising this question. This nanogel-based delivery system, created by crosslinking pullulan and an oxidatively degradable conjugated polymer PDDA, is inert to endogenous oxidative conditions, but can well degrade in the presence of ROS generated by photosensitizers upon light laser irradiation. This design provides the nanogels

enhanced stability in physiological conditions and allows them to cumulate efficiently in GBM lesions after intravenous administration. NIR light is then applied manually to activate the nanogels when they reach maximal accumulation in GBM lesions. The loaded ICG can generate ROS to deform the nanogels and trigger the burst release of all drugs for facilitated extravasation and deep tumor penetration. The efficient cell uptake of these nanogels by tumor cells was evidenced by the CLSM images (Fig. R9a, or Fig. 4b in the revised manuscript) and flow cytometry analysis (Fig. R9b, or Fig. 4c in the revised manuscript). In addition, we have also validated the deep tumor cell uptake using a U87MG 3D spherical model, which showed that NIR-activated ARNGs@TMZ/ICG displayed much stronger ICG fluorescence than other controls (Fig. R9c, or Fig. 4d in the revised manuscript). Moreover, the nanogels are camouflaged with erythrocyte membranes and decorated with a widely-accepted brain-targeting ligand ApoE, which warrants the prolonged blood circulation, efficient BBB crossing, and active tumor targeting of the nanogels. The efficient tumor cell uptake in the brain, together with the spatiotemporally controlled release of TMZ and ICG, allow for their sufficient accumulation in deep GBM lesions for the effective photodynamic-chemotherapy synergy.

Fig. R9 **a** CLSM images of U87MG cells receiving various treatments and stained by fluorescein 5-isothiocyanate (FITC) and 4',6-diamidino-2-phenylindole dihydrochloride (DAPI). Scale bar: 20 μm . **b** Flow cytometry analysis of U87MG cells receiving various treatments. **c** Uptake of ARNGs@TMZ/ICG by U87MG multicellular spheroids after 4 h of incubation with and without NIR irradiation. Z-stack imaging was progressed from the bottom into the core of the spheroids at an interval of 10 μm . Scale bar: 200 μm ; bottom left: Schematic

diagram of the 3D spherical U87MG model; bottom right: Quantification of the relative ICG mean fluorescence intensity (MFI, n = 3).

4. In Figure 3 d, e, after near-infrared light irradiation, why does the red blood cell membrane outside the hydrogel fall off, and what is the principle?

Response: We thank the reviewer for pointing out this issue. We would like to explain that the red blood cell membranes (RBCm) did not fall off from the nanogels after NIR irradiation. After NIR irradiation, the nanogels deformed due to the degradation of the skeleton PDDA, leaving the RBCm-wrapped nanoparticles softer than before. The RBCm, without the support of crosslinked nanogels, were more likely to fuse each other, as what we had observed from the TEM image (Fig. 3e). This result is also consistent with the increased particle size upon NIR irradiation as measured by DLS (Fig. 3f). In addition, we have retaken TEM images of the samples after the NIR irradiation, which further confirmed the fused morphology of irradiated nanoparticles.

Reviewer #3

General comments: This is an interesting article evaluating a novel nanoparticle formulation that releases drug cargo in response to NIR irradiation. Inclusion of an ApoE peptide targeting uptake across the BBB, this strategy is a promising and interesting strategy. Overall, the studies are clearly presented and appear carefully performed. However, there are several limitations that dampen enthusiasm for the manuscript that can be readily addressed.

1. Sole use of U87 is a limitation to the study. Confirming key studies in a second glioma cell line would be useful. Notably, patient-derived xenograft or glioma stem cell models are more current and applicable strategies.

Response: We really appreciate the reviewer for the elaborate review and very inspiring comments and suggestions. The reviewer's suggestion on key studies in a second glioma cell line is valuable and should greatly improve the quality of our work. As suggested, we have evaluated the anti-tumor effects of our developed nanogels in a glioma stem cells (GSCs) model. Accordingly, we have established a patient-derived GSCs CSC2 orthotopic mice model. The overexpression of low-density lipoprotein (LDL) receptors of CSC2 cells have been firstly verified to lay the fundamental of the active targeting of our design (Fig. R10, or Supplementary Fig. 7 in the revised manuscript). We next studied the cytotoxicity of NIR-activated ARNGs@TMZ/ICG on CSC2 cells, which efficiently inhibited the cell proliferation

with a considerably low cell viability (Fig. R11, or Supplementary Fig. 12 in the Supplementary Information).

Fig. R10 Expression levels of LDL receptor family, including LDL receptor (LDLR), LDLR-related proteins 1 and 2 (LRP1 and LRP2), in bEnd3 endothelial cells, U87MG GBM, GSCs CSC2, normal BV2 microglial cells and HA1800 astrocytes determined by Western blotting.

Fig. R11 Cell viability of GSCs CSC2 cells measured by CellTiter-Lumi™ luminescent cell viability assay at 48 h after receiving various treatments (n = 7). The incubation time with treatment agents: 4 h; NIR: 808 nm, 0.5 W cm⁻², 5 min; ICG concentration: 10 μg mL⁻¹; TMZ concentration: 10 μg mL⁻¹; Data are presented as mean ± SD (one-way ANOVA and Tukey's

multiple comparison test, *** $p < 0.001$).

We further validated the excellent anti-GBM efficacy of these nanogels in an orthotopic GSCs CSC2-bearing mice model (Fig. R12, or Fig. 7 in the revised manuscript). The treatment schedule was similar with that in U87MG mice model, injecting various formulations on Day 10, 12, 14, 16, and 18 after the tumor implantations, followed by NIR irradiation at 4 h after each injection for the mice receiving light treatment (Fig. R12a). The tumor bioluminescence images showed that the NIR-activated ARNGs@TMZ/ICG could effectively suppress the tumor growth during the treatment period. ARNGs@TMZ/ICG without irradiation also inhibited the tumor proliferation to some extent, but less efficient than with NIR activation (Fig. R12b). In sharp contrast, rapid tumor growth was observed for mice treated by PBS, with or without light irradiation. The quantitative bioluminescence intensity results were in line with the images and further confirmed the best anti-tumor efficacy of NIR-activated nanogels (Fig. R12c). The H&E staining of the whole brain confirmed that the mice treated by NIR-activated ARNGs@TMZ/ICG had the smallest tumor sizes (Fig. R12d). The mice in the experiment all maintained their body weights throughout the study (Fig. R12e). Importantly, NIR-activated ARNGs@TMZ/ICG markedly prolonged the mice survival with the median survival time of 63 d, which was significantly longer than mice receiving other treatments (Fig. R12f). The histological analysis of tumor slices showed that the NIR-activated nanogels induced the most tumor cell apoptosis and the least tumor cell proliferation (Fig. R12g). Collectively, the NIR-activated ARNGs@TMZ/ICG exhibited an anti-tumor effect in GSC CSC2 mice model as good as in U87MG mice models, demonstrating the excellent efficacy of our NIR-activatable biomimetic nanogels. Correspondingly, we have added an additional section in the revised manuscript on Page 21 (highlighted in red):

Efficient inhibition of orthotopic CSC2 glioblastoma stem cells (GSCs) tumors in mice.

The excellent anti-tumor effects of NIR-activated nanogels encouraged us to further investigate whether they could also restrain the incurable GSCs. Firstly, we established a patient-derived GSCs CSC2 orthotopic mice model that expressed luciferase stably (Fig. 7a, 7b). The overexpression of LDL receptors of CSC2 cells was verified to lay the fundamental of the efficient BBB crossing as well as the active internalization of our biomimetic nanogels (Supplementary Fig. 7). We next studied the cytotoxicity of NIR-activated ARNGs@TMZ/ICG on CSC2 cells, which efficiently inhibited the cell proliferation with a considerably low cell viability (Supplementary Fig.12). The tumor bioluminescence images showed that the NIR-

activated ARNGs@TMZ/ICG could effectively suppress the tumor growth during the treatment period. ARNGs@TMZ/ICG without irradiation also inhibited the tumor proliferation to some extent, but was less efficient than that with light treatment (Fig. 7b). In sharp contrast, rapid tumor growth was observed for mice treated with PBS, with or without light irradiation. The quantitative bioluminescence intensity results were in line with the images and further confirmed the best anti-tumor efficacy of NIR-activation nanogels (Fig. 7c). The H&E staining of whole brain further indicated that the NIR-activated ARNGs@TMZ/ICG treated mice had the smallest tumor size (Fig. 7d). Furthermore, the mice in the experiment all maintained their body weight throughout the study (Fig. 7e). Importantly, NIR-activated ARNGs@TMZ/ICG markedly prolonged the mice survival with a median survival time of 63 d, which was significantly longer than the mice receiving PBS (20 d) or nanogels without NIR (44 d, Fig. 7f). The histological analysis of tumor slices showed that the NIR-activated nanogels caused the most tumor cell apoptosis and the least tumor cell proliferation (Fig. 7g). Additionally, major organs in mice treated with NIR-activated ARNGs@TMZ/ICG showed no side effects (Supplementary Fig. 13). Collectively, the NIR-activated ARNGs@TMZ/ICG exhibited an anti-tumor effect in GSC CSC2 mice model as good as in U87MG mice models, demonstrating the excellent efficacy of our NIR-activatable biomimetic nanogels.

Fig. R12 Anti-tumor efficacy of NIR-activatable ARNGs@TMZ/ICG against orthotopic GBM stem cells (GSCs) mice. **a** Schematic illustration of the establishment of GSCs mice model. CSC2-Luc cells were orthotopically inoculated into the brains of 6-8 weeks nude mice. On Day 10 after the tumor inoculation, mice with a similar bioluminescence intensity were

selected and randomized into 4 groups (n = 5). Various formulations were intravenously injected at a dose of 10 mg TMZ equiv. kg⁻¹ and 10 mg ICG equiv. kg⁻¹ on Day 10, 12, 14, 16, and 18 post tumor implantation. **b** In vivo bioluminescence images of orthotopic GSCs in live mice receiving different treatments. **c** Quantified tumor bioluminescence levels of orthotopic GSCs in each group. **d** H&E-staining images of the orthotopic brain tumor tissues excised from the mice in each group. **e** Body-weight changes in mice. **f** Kaplan-Meier analysis of the mice. **g** TUNEL, γ H2AX, CC3 and Ki67 staining images of the orthotopic brain tumor tissues excised from the mice in each group. Scale bars: 200 μ m for TUNEL images and 60 μ m for γ H2AX, CC3 and Ki67 images. Data are presented as mean \pm SD (one-way ANOVA and Tukey multiple comparisons tests, * p < 0.05, ** p < 0.01, *** p < 0.001).

2. Figure 4F/G. Incubation with TMZ for 24h is insufficient to evaluate cytotoxicity specifically from TMZ, which occurs over 2-3 cell cycles. The concentration of TMZ in the various conditions should be specified.

Response: We really thank the reviewer for this very valuable comment. According to the suggestion, we have conducted the cell viability and apoptosis assay for a longer incubation time of 48 h. The results showed a similar profile with the assay incubated for 24 h. NIR-activated ARNGs@TMZ/ICG could kill the tumor cells efficiently as compared with other controls (Fig. R13a, or Fig. 4f in the revised manuscript). Notedly, the cell viability was 23% at 48 h after being treated with NIR-activated nanogels, which was significantly lower than that at 24 h post treatment (nearly 50%), confirming that the tumor cell suppression ability of these nanogels was closely related to incubation time. The apoptosis results from the flow cytometry analysis were in line with the cell viability study (Fig. R13b, or Fig. 4g in the revised manuscript). Accordingly, we have replaced the original figures with the newly obtained 48 h-treatment results.

In addition, the TMZ concentration in all the formulations was **10 μ g mL⁻¹**, which has been specified in the legends as well as methods section.

Fig. R13 a Cell viability of U87MG cells at 48 h after receiving various treatments (n = 5). **b**, Apoptosis analysis of U87MG cells by flow cytometry at 48 h after receiving various treatments and stained by PI and Annexin V. For all studies, incubation time with treatment agents: 4 h; NIR: 808 nm, 0.5 W cm⁻², 5 min; ICG concentration: 10 μg mL⁻¹; TMZ concentration: 10 μg mL⁻¹; Data are presented as mean ± SD (one-way ANOVA and Tukey's multiple comparison test, ****p*<0.001).

3. Tetrazolium salt-based assays, such as MTT, XTT MTS assays, are affected by the mitochondrial effects of TMZ and are not accurate assessment of TMZ toxicity. An orthogonal

assay should be used to confirm the results.

Response: Thanks for this very helpful suggestion. As suggested, we have used CellTiter-Lumi™ luminescent cell viability assay kit to confirm the cell viability of TMZ-loaded nanogels. The TMZ toxicity detected by this new assay displayed a similar profile as by MTT, in which NIR-activated ARNGs@TMZ/ICG resulted in the lowest cell viability among all groups. To keep the result precise and consistent, we have adopted this assay method in the cell toxicity study with 48 h incubation (Fig. 4f), synergy analysis (Supplementary Fig. 5), and the cell toxicity study of these nanogels towards GBM stem cells (Supplementary Fig. 12).

4. Cytotoxicity is associated with ICG alone and TMZ alone. Synergy analyses could be useful to define the impact of the combined nanoparticle simply reflects additive toxicities.

Response: We thank the reviewer for the nice comment. As suggested, we have performed the synergy analysis by studying the cytotoxicity of nanogels loaded with single drug and drug combination at various drug concentrations. It is shown that the combination index (CI) was below 0.5 when the concentrations of TMZ and ICG were within a range from 0.125 to 80 $\mu\text{g mL}^{-1}$ (Fig. R14, or Supplementary Fig. 6 in the Supplementary Information), demonstrating the synergistic effects of TMZ and ICG co-delivered via our nanogels. This result has been added in the revised manuscript on Page 12 (highlighted in red): **The combination index between PDT and chemotherapy were below 0.5 when TMZ and ICG concentrations were from 0.125 to 80 $\mu\text{g mL}^{-1}$ (Supplementary Fig. 6).**

Fig. R14 a Cell viability of U87MG GBM cells after being treated with ARNGs@TMZ/ICG+L,

ARNGs@ICG+L, and ARNGs@TMZ, respectively. (TMZ and ICG concentrations were equal, both ranging from 0.125 to 80 $\mu\text{g mL}^{-1}$) **b** The CI values of NIR-activated ARNGs@TMZ/ICG treatment at the corresponding TMZ/ICG concentrations. The incubation time with treatment agents: 48 h; NIR: 808 nm, 0.5 W cm^{-2} , 5 min; Data are presented as mean \pm SD (n=5).

5. The ApoE decoration for the nanoparticles could be quite important for BBB and tumor targeting. In this context, creating and testing nanoparticles with a scrambled peptide sequence but the same amino acid content is an important control that should be used in key experiments.

Response: We highly appreciate the reviewer for this very important and valuable comment. We agree with the reviewer that the ApoE decoration does play a key role in improving the BBB permeability and tumor targeting of the biomimetic nanogels. The reviewer's suggestion on scramble peptide is also essential. ApoE enhances the BBB permeability and the tumor cell active targeting of nanoparticles mainly through low density lipoprotein (LDL) receptor-mediated endocytosis and transcytosis, while LDL receptor family was overexpressed by both endothelial cells of BBB and brain tumor cells (U87MG and CSC2). The interaction between LDL and ApoE has been extensively investigated previously, using scrambled peptide sequence but the same amino acid residues and analogues of ApoE (e.g., *Biochemistry* **2000**, 39, 213-220; *J. Bio. Chem.* **2003**, 278, 48529; *Biochemistry* **2004**, 43, 7328-7335). Decorating nanoparticles with ApoE has been established as a reliable strategy to enhance BBB crossing and tumor targeting (*Adv. Drug. Deliv. Rev.* **2001**, 47, 65-81). More recently, we and others have achieved efficient BBB crossing and brain tumor active targeting using ApoE-decorated nanoparticles (*ACS Nano*, **2018**, 12, 11070-11079; *Adv. Ther.*, **2021**, 4, 2000092; *ACS Nano*, **2022**, 16, 6293-6308; *Exploration*, **2022**, 20210274). In this manuscript, we focused on the design of the nanogels to achieve spatiotemporal control of the drug release for deep tumor penetration. We needed to choose a well-established, reliable functionalization strategy of the nanogels to ensure efficient BBB crossing and tumor targeting. Therefore, we employed ApoE to decorate the nanogels, based on the literature and our previous work, so that the loaded TMZ and ICG could be released inside the brain as desired for distal tumor cell uptake. Nevertheless, we evidenced the enhanced BBB permeability and tumor uptake through the ApoE functionalization by our in vitro (Figure 4b, 4c, Supplementary Fig. 4) and in vivo (Figure 5c, 5d, 5d-5h) experiments in contrast with non-targeting controls. We hope our explanation could fully address the reviewer's concern.

6. Throughout the in vitro studies, the effective concentration of TMZ should be provided for both free and nanoparticle formulated drug treatments.

Response: The effective concentrations of TMZ and ICG were both $10 \mu\text{g mL}^{-1}$ for all formulations (nanogels and free drugs), which have been specified in the legends of corresponded figures and the methods section.

7. Figure 7 – the changes in ALT and AST are lower in the free TMZ treatment as compared to placebo or nano formulation. These changes in ALT and AST are subtle and unlikely to be clinically meaningful.

Response: We appreciate the reviewer's constructive comments. Considering the reference values of ALT and AST are both 0-40 U/L (may be fluctuated according to the gender, age, species, etc.), although several blood parameters of mice treated with free TMZ were lower than that of PBS treatment, the levels may still within the normal variation. To clarify, we have removed the following inappropriate description in the revised manuscript on Page 24: “In sharp contrast, the values of alanine aminotransferase (ALT) and aspartate aminotransferase (AST) of the mice treated by mixed solution of TMZ and ICG (Free TMZ/ICG) were significantly higher than those treated by PBS on Day 2 and 4 (Fig. 8e, 8f), implying that the free drugs caused hepatotoxicity”.

REVIEWERS' COMMENTS

Reviewer #1 (Remarks to the Author):

The authors have made adequate revisions according to the comments of reviewers. This paper can now be accepted as it is.

Reviewer #2 (Remarks to the Author):

The highlight of this study is that photodynamic therapy promotes the penetration of chemotherapy drugs into the tumor to kill tumor cells. ApoE achieves brain targeting and red blood cell coating achieves long circulation of the carrier *in vivo*, which combines the advantages of carrier design in recent years. The design of the article is reasonable, and the revision is relatively complete according to the comments of the reviewers. I agree to the publication of its article.

Reviewer #3 (Remarks to the Author):

The authors have thoroughly and convincingly addressed this reviewer's concerns.